# New Methodologies as Opportunities in the Study of Bacterial Biofilms, Including Food-Related Applications

**DOI:** 10.3390/microorganisms13051062

**Published:** 2025-05-02

**Authors:** Francesca Coppola, Florinda Fratianni, Vittorio Bianco, Zhe Wang, Michela Pellegrini, Raffaele Coppola, Filomena Nazzaro

**Affiliations:** 1Institute of Food Science, CNR-ISA, 83100 Avellino, Italy; fratianni@isa.cnr.it (F.F.); coppola@unimol.it (R.C.); 2Department of Agricultural Sciences, University of Naples “Federico II”, Piazza Carlo di Borbone 1, 80055 Portici, Italy; 3Institute of Applied Sciences and Intelligent Systems “Eduardo Caianiello”, Via Campi Flegrei, 80078 Pozzuoli, Italy; vittorio.bianco@isasi.cnr.it (V.B.); zhe.wang@isasi.cnr.it (Z.W.); 4Dipartimento di Ingegneria Chimica, Dei Materiali e della Produzione Industriale, University of Napoli Federico II, Piazzale Vincenzo Tecchio 80, 80125 Napoli, Italy; 5Department of Agricultural, Food, Environmental and Animal Science, University of Udine, Via Sondrio 2/a, 33100 Udine, Italy; 6DiAAA, University of Molise, Via De Sanctis s.n.c., 86100 Campobasso, Italy

**Keywords:** biofilm, DNA-methods, CRISPR, biospeckle, microscopy, organoids, microfluidics

## Abstract

Traditional food technologies, while essential, often face limitations in sensitivity, real-time detection, and adaptability to complex biological systems such as microbial biofilms. These constraints have created a growing demand for more advanced, precise, and non-invasive tools to ensure food safety and quality. In response to these challenges, cross-disciplinary technological integration has opened new opportunities for the food industry and public health, leveraging methods originally developed in other scientific fields. Although their industrial-scale implementation is still evolving, their application in research and pilot settings has already significantly improved our ability to detect and control biofilms, thereby strengthening food safety protocols. Advanced analytical techniques, the identification of pathogenic species and their virulence markers, and the screening of “natural” antimicrobial compounds can now be conceptualized as interconnected elements within a virtual framework centered on “food” and “biofilm”. In this short review, starting from the basic concepts of biofilm and associated microorganisms, we highlight a selection of emerging analytical approaches—from optical methods, microfluidics, Atomic Force Microscopy (AFM), and biospeckle techniques to molecular strategies like CRISPR, qPCR, and NGS, and the use of organoids. Initially conceived for biomedical and biotechnological applications, these tools have recently demonstrated their value in food science by enhancing our understanding of biofilm behavior and supporting the discovery of novel anti-biofilm strategies.

## 1. Introduction

The increasing demand for safe and high-quality food has driven the exploration and adoption of advanced analytical technologies. Traditional food safety and quality assessment methods, while effective, often suffer from limitations such as time consumption, high costs, and complexity. Consequently, emerging technologies initially developed for other scientific disciplines are now being repurposed to enhance food safety protocols. Among these, microfluidics, optical methods (such as Laser Confocal Scanning Microscopy), Atomic Force Microscopy (AFM), electron microscopy, biospeckle imaging, genetic methods, such as CRISPR, qPCR, and NGS, and organoids have gained considerable attention.

These methodologies enable precise detection of foodborne pathogens, biofilm formation, and novel antimicrobial substances, offering innovative solutions to food safety challenges. For example, Laser Confocal Scanning Microscopy allows non-invasive, real-time visualization of biofilm architecture and cell viability at different depths, while AFM provides nanomechanical data such as adhesion and elasticity, useful for understanding biofilm robustness and resistance. Microfluidic platforms simulate food-relevant environments and allow detailed study of biofilm heterogeneity, interspecies interactions, and antimicrobial response dynamics under controlled conditions.

From a molecular perspective, CRISPR technology enables targeted gene editing and interference (e.g., CRISPR-Cas9 and CRISPRi), facilitating the investigation of gene functions involved in biofilm formation, quorum sensing, and antimicrobial resistance. CRISPR-based biosensing systems are also being developed for the detection of specific biofilm-associated genes. However, challenges include efficient delivery systems for some microbial strains and managing off-target effects in complex microbiomes. Quantitative PCR (qPCR) remains a gold standard for detecting and quantifying biofilm-related genes and microbial populations due to its speed and sensitivity. Still, it requires prior knowledge of the target sequences and does not provide spatial or functional information about microbial communities. Next-generation sequencing (NGS), especially in its metagenomic and meta transcriptomic forms, allows comprehensive profiling of the taxonomic composition and metabolic activity within mono- or multispecies biofilms. Despite its depth, NGS demands advanced bioinformatic analysis and may not distinguish between live and dead cells without complementary techniques.

Finally, organoid models—3D cell cultures derived from animal or human tissues—are gaining traction for studying host–pathogen interactions in a physiologically relevant context. Though still in the early stages for food microbiology applications, they offer promising insights into how pathogens and biofilms interact with host barriers. Figure 1 briefly summarizes some foodborne and beneficial microorganisms and the methodologies useful that will be described in this review.

By leveraging the complementary strengths of these methods, researchers can obtain a multifaceted understanding of biofilm behavior, pathogen dynamics, and antimicrobial strategies—supporting the development of more effective and precise food safety interventions.

## 2. Microbial Biofilm

Microbial biofilms are highly structured communities of microorganisms that adhere to surfaces and are embedded within a self-produced extracellular polymeric substance (EPS) matrix. These complex microbial ecosystems can be easily seen as prevalent in natural, industrial, and clinical environments, and they can possess a double, positive, and negative valence, contributing thus to both beneficial and detrimental biological processes. The dual nature of microbial biofilms—as both beneficial (e.g., in fermentation or microbial stability) and harmful (e.g., spoilage or pathogenicity)—complicates their characterization and control in agro-industrial microbiomes. Their functional roles are context-dependent, making it difficult to establish universal markers or mitigation strategies. Interventions targeting harmful biofilms may unintentionally disrupt beneficial communities, while their dynamic and adaptive nature further challenges selective control. As such, effective management requires tailored, context-specific approaches that balance microbial ecology with food safety objectives. The capacity of microbes to form biofilms assumes significant importance in different fields of research, such as medicine, food safety, biotechnology, and environmental microbiology.

### 2.1. Biofilm Formation Process

Biofilm development occurs through a well-defined sequence of stages [1,2]:**Initial Attachment**: When the bacterial mass level exceeds a certain threshold, the still planktonic bacterial cells aggregate and, in the form of microcolonies, detach and colonize a specific surface. From a biophysical perspective, adhesion involves van der Waals forces and hydrophobic interactions [3].

Up to this point, it is still relatively easy to remove the adhered cells.

**The transformation from planktonic to sessile cells**, driven by changes in the cellular metabolic pathway, leads the bacterial cells to produce exopolysaccharides, proteins, and nucleic material. These components form a gelatinous and viscous substance that surrounds and protects the sessile cells, resembling a dome-like structure. Exopolysaccharides strengthen their adhesion and promote colonization, then proliferate and form microcolonies. Within this environment, they start to communicate via the mechanism of the so-called quorum sensing [4]. Quorum sensing (QS) plays a central role in orchestrating transcriptional regulation and phenotypic heterogeneity within maturing biofilm consortia. Through the production and detection of small signaling molecules (autoinducers), QS enables microbial populations to sense cell density and coordinate gene expression collectively. This regulation governs key biofilm-associated processes such as EPS production, motility, virulence factor expression, and stress responses. Importantly, QS also contributes to the spatial and functional diversification of cells within the biofilm, leading to phenotypic heterogeneity—an essential feature for biofilm resilience and adaptability. This heterogeneity results in distinct subpopulations with specialized roles (e.g., matrix producers, metabolically dormant cells, persisters), which complicates the effectiveness of antimicrobial interventions. Moreover, QS-mediated communication facilitates metabolic synchrony across the community, optimizing resource utilization and enhancing survival under fluctuating environmental conditions. Therefore, disrupting QS pathways represents a promising strategy to interfere with this synchrony, attenuate biofilm robustness, and sensitize microbial populations to antimicrobial agents or environmental stressors.**Maturation of biofilm and dispersion of sessile cells**.

The biofilm architecture develops into a three-dimensional structure with water channels for nutrient and waste transport [5]. At this point, some sessile cells detach from the biofilm to revert to a planktonic state, allowing the colonization of new surfaces [6].

Biofilms in the food industry contain diverse microbial communities, including pathogenic and spoilage bacteria. The biofilm matrix consists of different macromolecules, in particular: polysaccharides, essential for biofilm cohesion and adhesion [7]; proteins and enzymes, which aid in biofilm stability and resistance [8]; extracellular DNA (eDNA), which contributes to structural integrity and genetic exchange [9]; and lipids: helpful in enhancing the adhesion to hydrophobic surfaces [10]. In addition, microbial biofilm produces the so-called quorum sensing molecule (QSM) signaling compounds, such as acyl-homoserine lactones (AHLs) in Gram-negative bacteria and autoinducing peptides (AIPs) in Gram-positive bacteria, all capable of regulating biofilm development [11]. Finally, sessile cells within biofilms can produce, in turn, different metabolites and secondary compounds, such as siderophores and other metabolites, which can contribute to enhancing their survival and competitiveness [12].

The biofilm exhibits a stratified arrangement: a superficial layer, populated by metabolically active cells, with access to oxygen and nutrients; an intermediate layer, densely populated, with less metabolically active bacteria and a higher accumulation of EPS; and a deep layer characterized by cells in a quiescent or dormant state, with limited availability of oxygen and nutrients. Water channels, essential structures for nutrient distribution and the removal of catabolites within the microbial community, are also present.

Figure 1 shows the different stages, from the formation of the biofilm to the release, from the mature biofilm, of new biofilm micro-niches that will colonize new surfaces.

The process can be resumed in the following steps:

When the bacteria reach a certain growth, they start attaching to a surface through physical forces and cellular appendages (a).

The adhesion process is reversible. In the second phase (b), we observe an improvement in the attachment, also due to the production by the bacteria of adhesins and gene upregulation.

From the large bacterial formation, some microcolonies (c) form and detach.

These microcolonies are covered with a composite matrix made of protein, polysaccharide, and nucleic material (d). Additionally, a cell communication system begins, mediated by the quorum sensing system, which also plays a role in the biofilm maturation process. The matrix protects the cells from antibiotic attacks and immune cells. The cells within these domes remain in a state of nutritional dormancy and gradually change their biochemical characteristics, making them more virulent compared to their planktonic bacterial counterparts.

However, when toxic catabolites accumulate within the exopolysaccharide matrix and the availability of nutrients falls below a certain minimum threshold, the exopolysaccharide matrix opens, leading to the release of cells that, in the form of microcolonies, colonize other surfaces (e).

### 2.2. Biofilm Resistance Mechanisms

The multidrug resistance phenotype observed in biofilm-embedded bacteria is supported by a combination of molecular and structural adaptations. One key mechanism is the upregulation of efflux pump systems, which actively expel a broad range of antimicrobial agents and are often more highly expressed in biofilm-associated cells than in their planktonic counterparts. Additionally, the biofilm matrix itself—formed by the exopolysaccharides—acts as a diffusion barrier, limiting antibiotic penetration and creating gradients of nutrients and oxygen that contribute to metabolic quiescence in deeper biofilm layers. This quiescent state renders cells, mainly those in the deeper layers of biofilms, less susceptible to antibiotics that target actively dividing cells. Some bacterial cells within biofilms defend themselves against antibiotic attacks by expressing multidrug efflux pumps that expel toxic compounds. Moreover, biofilms facilitate horizontal gene transfer through increased cell density, close physical proximity, and the presence of extracellular DNA within the matrix, which not only contributes structurally but also serves as a genetic reservoir. Additionally, microbial defense is carried out through the so-called genetic adaptation. In fact, horizontal gene transfer and mutation rates are increased within biofilms, enhancing adaptive resistance. All these conditions enhance the rates of conjugation, transformation, and transduction, promoting the dissemination of resistance genes. Together, these adaptations create a highly resilient microbial community, complicating eradication efforts and contributing to the persistence of multidrug-resistant infections in clinical and industrial settings [13,14,15,16,17,18]. To address these challenges, it is essential to explore strategies that target biofilm-specific resistance mechanisms. For example, disrupting the EPS matrix using enzymatic treatments (such as DNases, proteases, or polysaccharide-degrading enzymes) can enhance the penetration and efficacy of conventional disinfectants. Inhibiting quorum-sensing systems through quorum-quenching agents can block cell-to-cell communication pathways crucial for coordinated defense responses. Moreover, the use of adjuvants or metabolic stimulators may help sensitize dormant persister cells to antimicrobial agents. Emerging tools like CRISPR interference (CRISPRi) offer precision-targeted gene silencing of biofilm-related pathways, including those regulating adhesion, matrix production, or resistance. Integrating these approaches into cleaning and sanitation protocols may significantly reduce biofilm persistence on food-contact surfaces, minimize cross-contamination risks, and strengthen overall food safety and hygiene practices.

## 3. Foodborne Pathogens Associated with Biofilms

Biofilms serve as reservoirs for foodborne pathogens, increasing the risk of contamination and foodborne diseases. Common biofilm-forming pathogens include the following:

*Listeria monocytogenes*, a Gram-positive, facultatively anaerobic bacterium, is widely known as the causative agent of listeriosis. This severe infection primarily affects pregnant women, neonates, the elderly, and individuals with compromised immune systems. *L. monocytogenes* is a ubiquitous pathogen commonly found in environmental sources such as soil, water, and decaying plant matter and can contaminate a wide variety of foods, including dairy products, ready-to-eat meats, and vegetables. *Listeria monocytogenes* indeed poses a significant challenge in refrigerated food environments due to its unique physiological traits. As a facultative psychrotrophic microorganism, *L. monocytogenes* can not only survive but actively grow at refrigeration temperatures (as low as 0–4 °C), which significantly undermines the effectiveness of cold-chain protocols that are primarily designed to inhibit the growth of most foodborne pathogens.

One of the significant factors contributing to the persistence and pathogenicity of *L. monocytogenes* in food processing environments is its ability to form biofilms. Biofilm formation by *L. monocytogenes* in food processing environments represents a significant challenge for food safety, and plant extracts and essential oils can aid in fighting it [19]. Biofilms are typically formed on surfaces in contact with food, such as equipment, countertops, pipes, and storage containers. Biofilm adheres to stainless steel and food contact surfaces, persisting in dairy and meat processing plants [20]. Bacteria can continue to grow in the biofilm and may persist for long periods, even under sanitation conditions, thereby increasing the likelihood of contamination of food products. The formation of biofilms by *L. monocytogenes* is influenced by various factors, including the type of surface, environmental conditions (like temperature, humidity, and nutrient availability), and the presence of other microorganisms. Biofilms can act as reservoirs for *L. monocytogenes*, continuously releasing viable bacteria into the environment, which may contaminate food during processing, packaging, or storage. Biofilm-associated *L. monocytogenes* may demonstrate altered gene expression and phenotypic traits compared to planktonic (free-floating) bacteria, making them more virulent and difficult to eradicate [21]. These traits allow *L. monocytogenes* to subvert standard cold-chain decontamination protocols, persist in processing facilities, and cross-contaminate ready-to-eat products. These challenges underscore the need for more targeted sanitation strategies, including the disruption of biofilms and the implementation of more stringent monitoring programs, particularly in cold environments.

*Salmonella* spp. causes foodborne illnesses worldwide and a wide range of symptoms, including gastrointestinal distress, fever, and, in some severe cases, bloodstream infections. *Salmonella* infections are primarily associated with consuming contaminated food, particularly poultry, eggs, and dairy products. The persistence and antimicrobial resistance of *Salmonella* biofilms are indeed the result of a complex interplay between environmental factors—such as the physical-chemical properties of food-contact substrates—and intrinsic bacterial regulatory mechanisms. Food-contact surfaces like stainless steel, plastic, or rubber vary in surface energy, hydrophobicity, roughness, and charge—all of which influence bacterial adhesion and subsequent biofilm formation. For example, rougher or more hydrophobic surfaces can enhance initial attachment by providing shelter from shear forces and facilitating microcolony establishment. These surface characteristics can also influence nutrient retention and microenvironment stability, creating favorable conditions for biofilm maturation. On the microbial side, *Salmonella* spp. activates a range of intracellular signaling pathways in response to surface contact and environmental cues. Key among these is the cyclic-di-GMP signaling system, which regulates the transition from a planktonic to a sessile lifestyle and promotes the production of extracellular polymeric substances (EPS). Additionally, quorum sensing and stress response regulators (e.g., RpoS, CsgD) orchestrate the expression of biofilm-associated genes, enhancing tolerance to desiccation, oxidative stress, and antimicrobials.

The synergy between surface properties and intracellular responses leads to the development of structured, resilient biofilms that are notoriously difficult to eradicate. This dual mechanism emphasizes the importance of designing anti-fouling surfaces and targeting biofilm-specific regulatory pathways to effectively control *Salmonella* contamination in food-processing environments.

The bacteria can survive in diverse environments, including human and animal intestines and the food processing environment, making it a significant concern in food safety. In *Salmonella*, biofilm formation typically occurs on solid surfaces such as stainless steel, plastic, and glass, which is common in food processing and storage environments. *Salmonella* adheres to surfaces via pili and other adhesive structures. The bacteria then secrete extracellular polymeric substances (EPS) that encapsulate them, forming the biofilm matrix. This matrix provides structural stability, protects the bacteria from external stressors, and facilitates nutrient exchange among bacterial cells. The mature biofilm can harbor large populations of bacteria, making eradication difficult [22]. The presence of biofilms in food processing environments is a critical concern for food safety. *Salmonella* biofilms can persist on surfaces even after cleaning and disinfection procedures, posing a significant risk of contamination. The resistance of biofilms to antimicrobial agents and physical removal methods means that *Salmonella* can continue to survive and proliferate on food-contact surfaces, leading to recurrent contamination of food products. *Salmonella* biofilms are particularly resilient in environments with low humidity, fluctuating temperatures, and the presence of organic matter [23]. These factors are common in food processing facilities where equipment and surfaces are often exposed to residual food particles. Biofilm formation on equipment such as conveyors, cutting boards, and food-processing machinery can serve as a reservoir for pathogens, facilitating cross-contamination during food production.

*Escherichia coli* is usually found in the intestines of warm-blooded organisms, including humans and animals, where it forms part of the normal gut flora. Most strains of *E. coli* are harmless, but certain pathogenic strains, such as *E. coli* O157:H7, are associated with severe foodborne illnesses, including hemolytic uremic syndrome (HUS), diarrhea, and kidney failure [24]. The pathogenic strains of *E. coli* can cause widespread outbreaks, particularly through contaminated food such as raw ground meat, unpasteurized milk, and fresh produce [25]. *E. coli* is a major pathogen in the food industry, and its ability to form biofilms significantly contributes to its persistence in food processing environments. Biofilm formation in *E. coli* has significantly contributed to its persistence and pathogenicity, especially in food processing environments. In the food industry, *E. coli* biofilm formation on food-contact surfaces can increase the risk of contamination and its transmission to food products, leading to significant public health concerns. Biofilm formation in *E. coli* provides numerous advantages, including enhanced resistance to antimicrobial agents and increased survival in harsh environmental conditions. The biofilm’s three-dimensional architecture provides a physical barrier that limits the penetration of biocidal agents, particularly oxidative disinfectants such as hydrogen peroxide or peracetic acid. The extracellular matrix—also containing colanic acid, proteins (e.g., curli fimbriae), and extracellular DNA—acts as a protective scaffold that can neutralize or sequester reactive oxygen species before they reach the deeper layers of the biofilm.

Moreover, this matrix creates steep chemical gradients, leading to reduced metabolic activity in the biofilm’s inner regions. These metabolically quiescent cells are less susceptible to oxidative stress and contribute to the biofilm’s persistence. The structural integrity of the matrix also absorbs mechanical shear forces, enabling *E. coli* biofilms to resist removal by physical cleaning methods commonly applied in food-processing facilities.

Together, the matrix composition and biofilm architecture create a highly protective niche, promoting tolerance to both chemical and mechanical interventions. This highlights the need for integrated cleaning strategies that combine matrix-disrupting agents with biocides and mechanical action to ensure effective biofilm eradication.

In the context of the food industry, biofilm-forming *E. coli* strains are especially concerning because they can persist on food-contact surfaces for extended periods, complicating sanitation efforts [26]. The biofilm matrix also acts as a reservoir for *E. coli*, facilitating the spread of contamination through cross-contamination in food processing environments [27]. *E. coli* biofilms are of particular concern because they can form on equipment such as stainless-steel surfaces, conveyor belts, and cutting tools, where they serve as persistent reservoirs of bacteria that can contaminate food products [28]. In food processing environments, *E. coli* biofilms can lead to the continuous contamination of food items, even after repeated cleaning and sanitation efforts. Studies have shown that biofilms of *E. coli* can survive on surfaces for days to weeks, even in the presence of cleaning agents, making them a significant challenge in controlling contamination [29]. The biofilm-forming ability of *E. coli* increases the likelihood of contamination in various food products, including meat, dairy, fruits, and vegetables. Raw and fermented meats are common vehicles for pathogenic *E. coli*, and biofilm formation on equipment used in meat processing can lead to persistent contamination even in well-managed facilities [30,31]. Additionally, its biofilms in dairy processing equipment can serve as a source of contamination for milk and dairy products [32]. Biofilms contribute to the virulence and pathogenicity of *E. coli* by protecting the bacteria from both environmental stress agents and host immune responses and inhibiting the effectiveness of antibiotics and disinfectants; it means that *E. coli* cells within the biofilm are often more resistant to treatment compared to planktonic cells [13]. Furthermore, biofilms may facilitate the transmission of *E. coli* by contaminating food-contact surfaces, leading to foodborne illness outbreaks. The persistence of *E. coli* biofilms in food-processing environments raises concerns about their role in outbreaks and the difficulty of eradicating these biofilms from food-contact surfaces once established [33].

*Staphylococcus aureus* can cause various infections, including pneumonia, endocarditis, and septicemia. In the food context, *S. aureus* is a significant concern due to its potential to cause foodborne illnesses. It can produce heat-stable enterotoxins that can lead to food poisoning, particularly in improperly handled or stored foods. This microorganism has a well-documented ability to form biofilms, which poses significant risks to food safety, as it can be challenging to remove through standard cleaning procedures [34]. In the food industry, biofilms can form on various surfaces, such as food processing equipment, storage containers, and surfaces that contact food. Once *S. aureus* establishes itself on these surfaces, it can create a reservoir of bacteria that continues to pose a risk of contamination. The biofilm matrix protects the bacteria from antimicrobial treatments, making it more challenging to eliminate compared to planktonic [free-floating] bacteria [35,36]. This resistance to cleaning protocols increases the risk of cross-contamination and long-term contamination of food products. The formation of biofilms by *S. aureus* in food environments involves several factors, including the expression of surface proteins, such as clumping factors (Clf) and fibronectin-binding proteins (FnBPs), which mediate adhesion to surfaces. Additionally, polysaccharide intercellular adhesin (PIA) production is critical for biofilm matrix formation and stability. PIA is responsible for the aggregation of bacterial cells, contributing to the structural integrity of the biofilm. Central to *S. aureus* biofilm formation is the accessory gene regulator (agr) quorum-sensing system, which modulates the expression of virulence factors in a density-dependent manner. During the initial stages of biofilm development, agr activity is downregulated, promoting surface attachment and accumulation. As the biofilm matures, agr is reactivated, facilitating the expression of proteases and toxins that aid in biofilm dispersal and host colonization. The SarA (staphylococcal accessory regulator) and σ^B (alternative sigma factor) regulatory systems also play critical roles in promoting biofilm formation by repressing protease expression and enhancing the production of adhesins and extracellular polymeric substances. Key adhesins involved in *S. aureus* biofilm development include MSCRAMMs (Microbial Surface Components Recognizing Adhesive Matrix Molecules), such as fibronectin-binding proteins (FnBPs), clumping factors (ClfA, ClfB), and protein A, which facilitate strong adherence to both biotic and abiotic surfaces. Additionally, the polysaccharide intercellular adhesin (PIA), synthesized by the icaADBC operon, is essential for intercellular aggregation and biofilm maturation. Mechanistically, these factors not only enhance persistence on food-contact surfaces but also promote the accumulation of cells capable of producing enterotoxins, such as staphylococcal enterotoxin A (SEA), which are highly stable and remain active even after heat processing. Thus, the regulatory and adhesin systems that support biofilm development also contribute indirectly—but significantly—to foodborne toxigenesis by enabling the survival and proliferation of enterotoxigenic *S. aureus* populations in food-processing environments [37]. Biofilm formation by *S. aureus* in food processing environments is a significant concern due to its association with persistent contamination and the risk of foodborne illness outbreaks. Biofilms allow *S. aureus* to survive on surfaces even after cleaning and sanitation procedures, leading to potential contamination of food products. This allows the bacteria to produce enterotoxins that cause food poisoning, with symptoms such as vomiting, diarrhea, and abdominal cramps [38]. In addition to food safety implications, biofilm formation can also affect the shelf life and quality of food products, which could represent a reservoir for *S. aureus* to form complex communities with undesirable bacteria in multispecies biofilms [39].

*Pseudomonas* spp. are usually environmental opportunistics bacteria found in soil, water, and plant surfaces. Some *Pseudomonas* species are also important pathogens in human and animal infections, and foods [40,41], though many are non-pathogenic. Notably, *P. aeruginosa* is a significant pathogen, especially in immunocompromised individuals, causing infections such as pneumonia and wound infections [42,43]. However, species such as *P. fluorescens* and *P. putida* are more commonly associated with food spoilage and have a significant presence in the food industry. In food safety, *Pseudomonas* spp. are of particular concern due to their ability to form biofilms on food-contact surfaces, contributing to their persistence in food processing environments and complicating sanitation efforts. These biofilms increase the bacterium’s resistance to disinfectants and antibiotics and enhance their ability to contaminate food products. Biofilm formation in *the Pseudomonas* species is a complex and highly regulated process essential for survival in adverse conditions. Within biofilms, *Pseudomonas* exhibits heightened metabolic activity, especially in the production of extracellular enzymes such as proteases, lipases, and lecithinases, which catalyze the degradation of proteins, fats, and other structural components of food matrices. This enzymatic degradation not only compromises the sensory and nutritional quality of food products—manifesting as off-flavour, texture changes, and discoloration—but also facilitates nutrient release, which in turn supports further microbial growth and biofilm maturation. Notably, these processes are tightly regulated by quorum-sensing systems, such as the N-acyl-homoserine lactone (AHL)-mediated signaling pathways, which coordinate the expression of spoilage-related enzymes and secondary metabolites in a cell density-dependent manner. The intersection between biofilm formation, enzymatic spoilage, and quorum-regulated metabolite production contributes to a self-sustaining spoilage ecosystem. This poses a major challenge in industrial food processing environments, where persistent *Pseudomonas* biofilms can lead to recurrent contamination and substantial economic losses due to product recalls, reduced shelf life, and increased cleaning and downtime costs. Altogether, the enzymatic and regulatory sophistication of *Pseudomonas* spp. in biofilms underscores their central role in food spoilage and highlights the need for targeted interventions that disrupt quorum sensing and biofilm resilience in industrial settings [44]. The *Pseudomonas* biofilms can also exhibit heterogeneity, with some areas of the biofilm harboring more active cells while other parts are more dormant or resistant to antimicrobial treatments [45]. Various environmental factors, including nutrient availability, temperature, pH, and the presence of other microorganisms, influence biofilm formation in *Pseudomonas*. The ability of *Pseudomonas* to form biofilms on a wide range of surfaces, including stainless steel, plastics, and glass, makes it a significant concern in food processing and storage environments. *Pseudomonas* is widely associated with spoilage in the food industry, particularly in chilled foods like meat, fish, and dairy products. Their biofilm-forming ability plays a significant role in their persistence in food-processing environments, contributing to the continued contamination of food products. Maintaining hygiene and preventing cross-contamination in food production makes it more challenging. Biofilm-forming *Pseudomonas* species, especially *P. fluorescens* and *P. putida*, contribute to the spoilage of refrigerated foods by producing enzymes that degrade proteins, lipids, and other components of the food matrix, and that lead to off-flavors, discoloration, and a reduced shelf life [46]. *Pseudomonas* can also produce volatile compounds, including sulfur-containing compounds, which concur with the characteristic spoilage odors found in spoiled meat and fish [47]. Additionally, *Pseudomonas* biofilms are difficult to remove through standard cleaning methods or the use of conventional antibacterial agents due to their resistance to disinfectants and the protective role of the biofilm matrix [48]. It means that, after routine cleaning and disinfection, *Pseudomonas* biofilms can survive on food-contact surfaces, contributing to repeated contamination of food products. This represents a serious challenge for the food industry in ensuring food safety and preventing the growth of spoilage microorganisms. The economic burden of biofilm formation and resistance is considerable, with annual healthcare costs estimated in the billions. This emphasizes the urgent need for effective management strategies and innovative therapeutic solutions to improve patient outcomes.

Figure 2 shows the global annual economic burden (in billions) of different market sectors with biofilm-associated technologies, where the food and agriculture sector is the third.

Notably, mechanical and civil engineering bear the highest economic impact, with an estimated 2694 million USD, likely due to biofilm-induced corrosion, clogging, and material degradation in infrastructure and pipelines. This is followed by the medical and human health sector (386.8 million USD), where biofilms contribute to chronic infections and increased healthcare costs, particularly in relation to implants and catheters. Importantly, the food and agriculture sector ranks third, with a significant impact of 324 million USD. This underscores the challenges posed by biofilm formation on food-contact surfaces, equipment, and processing environments, where biofilms can lead to product spoilage, reduced shelf-life, contamination with pathogens, and increased cleaning costs. Other sectors such as homecare, water and waste treatment, and personal care also show considerable economic implications, reflecting how biofilms affect daily use products and systems. Interestingly, even areas such as oral care, marine environments, and energy and waste—though lower in relative cost—still incur measurable losses, reinforcing the cross-sectoral importance of biofilm management. This visual highlights the urgent need for innovative, sector-specific biofilm control strategies, particularly in food and agriculture, where balancing safety, product integrity, and sustainability remains a key priority. Although the image highlights the substantial economic losses caused by biofilms, particularly in engineering, healthcare, and food-related sectors, it is important to note that effective biofilm mitigation strategies often require only a fraction of these costs. For example, studies have shown that the implementation of targeted cleaning protocols, enzymatic treatments, or anti-biofilm coatings can reduce contamination and maintenance expenses by up to 50–80% in some sectors. In the food industry, the cost of regular sanitation using advanced enzymatic agents or quorum-sensing inhibitors may represent a minor percentage of overall operational expenses, yet it can prevent losses associated with product recalls, spoilage, or safety incidents, which can run into millions annually. Similarly, in medical settings, investment in anti-biofilm surface technologies or controlled-release antimicrobial coatings for devices has been shown to be highly cost-effective compared to the treatment of biofilm-associated infections. Despite this, the adoption of such technologies remains limited by a lack of awareness, regulatory inertia, and short-term cost constraints. A cost–benefit approach, incorporating long-term savings and risk reduction, is therefore essential when evaluating the implementation of biofilm control measures. In this context, the economic data presented in the image strongly support proactive investment in prevention and monitoring, particularly in critical areas like food production, where safety and quality are directly impacted [51].

### Microbial Biofilm and Food Safety

As indicated above, foodborne illnesses are attributed mainly to pathogenic bacteria and the biofilms they produce, representing a significant threat to public health and the food industry. Between 40% and 80% of microorganisms on the Earth are capable of forming biofilms [52]. In food-related environments, biofilms can form rapidly due to the presence of diverse microbial communities that thrive on food residues and infrastructure surfaces [53]. These biofilms often develop on materials commonly used in food processing, such as stainless steel, rubber, glass, polyethylene, polypropylene, and wood, many of which come into direct contact with food products [54]. Bacteria within these biofilms are frequently linked to human infections and contribute significantly to the persistence and spread of foodborne diseases. Studies highlight the critical role biofilms play in microbial contamination within food production facilities, whether they serve as reservoirs for pathogenic bacteria or contribute to cross-contamination events [55]. Biofilms are responsible for approximately 60% of foodborne outbreaks worldwide, making their presence in food industry settings a significant concern for food safety [56]. Their formation by foodborne pathogens is particularly problematic as it compromises product quality and endangers consumers [57,58]. Microbial colonization on surfaces in food manufacturing environments is strongly influenced by the physicochemical properties of both the microbial cell surface and the substrate, particularly surface hydrophobicity, charge, and roughness. Hydrophobic surfaces (e.g., certain plastics like polystyrene or polypropylene) generally promote stronger initial bacterial adhesion due to favorable hydrophobic interactions with microbial cell envelopes, which are often rich in lipids and hydrophobic proteins. In contrast, hydrophilic surfaces (e.g., glass or stainless steel) may support weaker initial adhesion; however, some bacteria can still attach efficiently by producing surface adhesins or extracellular polymeric substances (EPS) that overcome repulsive forces. Electrostatic interactions also play a role, with most bacterial cells carrying a net negative surface charge. This can lead to repulsion from negatively charged surfaces, unless mediated by divalent cations or conditioning films (e.g., proteins or organic residues), which can alter surface properties and facilitate attachment. Once established, these initial interactions influence the architecture, cohesion, and resilience of the developing biofilm. Biofilms on hydrophobic surfaces often exhibit tighter adhesion and more compact EPS matrices, contributing to enhanced resistance against shear stress and chemical disinfectants. Conversely, biofilms on hydrophilic surfaces may be more susceptible to detachment under mechanical cleaning, although some strains can still form robust, multilayered structures depending on environmental conditions and surface conditioning. Overall, the interplay between surface characteristics and microbial surface properties critically shapes biofilm behavior, influencing not only initial colonization but also long-term persistence, detachment, and resilience in food processing settings [59]. The dairy industry also faces specific challenges related to biofilm-forming bacteria. Thermophilic species like *Geobacillus* spp., capable of withstanding high temperatures (e.g., 65 °C), can survive pasteurization and affect powdered milk production [60]. On the other end of the temperature spectrum, psychrophilic bacteria like *Pseudomonas* form biofilms in cold environments, such as on the inner walls of milk storage tanks and pipelines. These bacteria produce enzymes that degrade milk components and reduce shelf life by compromising lipid stability [61]. Given the global impact of foodborne pathogens and the resilience of biofilms, there is an urgent need to develop innovative strategies for their prevention and control. Ensuring food safety requires continuous monitoring and proactive measures in industrial and clinical settings [62]. Prioritizing the elimination of harmful biofilms is essential for protecting public health and maintaining food quality across the entire production chain. However, it is equally important to recognize that even biofilms formed by technologically beneficial microorganisms, such as starter cultures or probiotics, can, under certain conditions, pose risks. If not properly monitored or controlled, these biofilms may act as reservoirs for spoilage organisms or even facilitate cross-contamination, especially in complex or mixed-species environments commonly found in dairy and meat processing facilities. Factors such as temperature abuse, inadequate cleaning protocols, or prolonged contact with surfaces can lead to a shift in microbial composition, allowing opportunistic or spoilage microbes to coexist within or adjacent to beneficial biofilms. To reduce these risks, it is critical to adopt strain-specific selection, ensure routine biofilm characterization, and apply precision hygiene practices that eliminate unwanted microorganisms without compromising the functionality of beneficial microbial populations.

## 4. Biofilm of Lactic Acid Bacteria

Lactic acid bacteria (LAB), including *Lactobacillus*, *Lactococcus*, and *Enterococcus* species, can form biofilms on various food contact surfaces, including stainless steel, plastic, and glass [63], which can have significant implications—both beneficial and detrimental—for the food industry [64,65,66,67].

The beneficial aspects of LAB biofilms include the following:

(1) Protective biofilms for food fermentation: LAB biofilms can be harnessed to improve food fermentation processes. For example, biofilm-associated LAB was demonstrated to be related to malolactic fermentation [68]; some LAB biofilms enhance probiotic properties, improving survival rates of beneficial bacteria in the gastrointestinal tract [63].

(2) Antimicrobial effects against pathogens: LAB biofilms can inhibit foodborne pathogens, such as *L. monocytogenes*, *Salmonella*, and *E. coli*, by producing organic acids, bacteriocins, and hydrogen peroxide and through the exclusion of the surface niches and nutrients. The metabolic products form part of the LAB secretome and contribute to the stabilization of fermented food microbiota, enhancing product safety and shelf life [69]. These beneficial effects are supported by several experimental studies. For example, *Lactobacillus plantarum* biofilms used in the fermentation of table olives showed enhanced resistance to environmental stresses and promoted desirable acidification profiles, contributing to improved product stability and sensory quality [70,71]. These examples illustrate the functional relevance of LAB biofilms in real food systems and support their application in improving fermentation outcomes and microbiological safety.

However, LAB biofilms can also have deleterious effects, particularly when they harbor spoilage-capable strains or when metabolic byproducts accumulate excessively. For example, excessive acidification or the overproduction of exopolysaccharides (EPS) can negatively impact texture, taste, or appearance. Additionally, certain LAB strains may facilitate the formation of off-flavors or biogenic amines under specific conditions, which are undesirable from a sensory and safety standpoint.

Furthermore, LAB biofilms can act as reservoirs for spoilage organisms and even harbor pathogenic bacteria under specific conditions, leading to cross-contamination [72]. The LAB metabolic secretome also plays a critical role in shaping interspecies competitiveness. By modulating environmental pH, redox potential, and nutrient availability, LAB biofilms can selectively inhibit or promote the growth of co-existing microbial species.

We can also consider biofilm-associated spoilage: certain LAB species, such as *Lactobacillus brevis* and *Leuconostoc mesenteroides*, contribute to spoilage in beer and dairy products by producing off-flavors and undesired texture changes [73]; biofilms on food-processing equipment can lead to persistent spoilage issues that are difficult to eliminate with conventional cleaning methods. This ecological modulation is central to both fermentation outcomes and spoilage dynamics, as shifts in community structure can favor opportunistic spoilage organisms if bioprotective LAB populations are destabilized. Overall, the dual role of LAB biofilms in fermented foods underscores the need for strain-specific selection and process optimization to ensure desirable microbial equilibria and prevent spoilage-related issues.

## 5. Biofilm of Fungi and Yeasts

Fungi display a remarkable ability to adhere to and colonize a wide range of surfaces, from living tissues and food products to industrial materials and rocks [74,75]. Much of what we know about fungal biofilms comes from their clinical relevance; genera like *Aspergillus*, *Candida*, *Coccidioides*, *Cryptococcus*, and *Pneumocystis*, are implicated in several human infections [76]. Structurally, fungal and yeast biofilms, such as those formed by *Candida* spp. or *Saccharomyces cerevisiae*, exhibit more complex architectures with hyphal elements (in filamentous fungi), a thicker matrix, and a distinct stratification of metabolic activity. These differences affect how antimicrobial agents penetrate and act within the biofilm. Therefore, while bacterial biofilms, functionally, often display coordinated behavior via quorum sensing systems (e.g., acyl-homoserine lactones or autoinducing peptides), regulating genes involved in resistance and biofilm maintenance, fungal biofilms utilize similar but distinct regulatory networks, including pathways like cAMP-PKA or MAPK, which govern morphogenesis and stress responses. These divergent signaling mechanisms contribute to the differential tolerance observed—fungal/yeast biofilms are often more recalcitrant to antifungals, particularly due to dense matrices and efflux pump activity, while bacterial biofilms may exhibit variable susceptibility depending on species and matrix composition [77,78]. In fermented food systems, both bacterial and yeast biofilms can contribute positively by stabilizing microbial communities, producing flavor compounds, and preventing pathogen colonization. However, under uncontrolled conditions, they can also facilitate spoilage—bacterial biofilms may lead to off-flavors, gas production, or proteolysis, while fungal biofilms can cause surface discoloration, off-flavours, and undesirable texture changes. Beneficial yeast biofilms also play a crucial role in food processing, especially *Saccharomyces cerevisiae*, which plays a crucial role in the production of traditional fermented foods. For instance, in wine fermentation, biofilm-forming strains of *S. cerevisiae* have been shown to improve flavor complexity and stability due to their prolonged metabolic activity at the air-liquid interface [79]. Biofilm initiation begins with yeast cells adhering to each other and surfaces, facilitated by adhesins [80] and hydrophobic/amyloid-like interactions [81], with the involvement of some proteins, such as Flo11p, Hsp12p, and Ccw14p, which are involved [82]. Following attachment, yeast cells can form hyphae or pseudo-hyphae and generate extracellular matrices [ECM], composed of polysaccharides, lipids, proteins, and nuclear material, which enhances surface adhesion [83,84] and offers structural support, nutrient reservoirs, and protection [85]. Yeasts are frequent in several foods, including wine [86], beer [87], dairy [88], and fruit [89]. Spoilage yeasts form persistent biofilms [90,91]. In table olives, co-aggregated biofilms of *S. cerevisiae* and LAB have been reported to enhance fermentation kinetics and inhibit spoilage microorganisms [92]. In dairy plants, biofilms in floor drains or membranes can harbor spoilage and pathogenic species [93]. Additionally, *Kluyveromyces marxianus* biofilms have demonstrated probiotic potential and thermotolerance, making them promising candidates for functional food applications and high-temperature processing [94]. Thus, the structural and functional divergence between these microbial groups necessitates tailored antimicrobial strategies and process controls, especially in complex food fermentations where both beneficial and spoilage-prone biofilms coexist.

## 6. Detection Methods to Investigate Biofilms in Food Environments

The study of bacterial biofilms in the food industry is essential for ensuring food safety, as biofilms can lead to persistent contamination in food processing and storage equipment. Detecting and analyzing biofilms is essential for timely intervention and infection management. While traditional methods are often effective, they typically do not identify the specific microbial species present within biofilms. Consequently, there is a growing demand for more advanced, cost-effective, and efficient methods to accurately evaluate the presence and characteristics. In recent years, several innovative technologies have emerged, providing advanced approaches for monitoring, analyzing, and preventing biofilm formation in the food sector. Below are some of the most promising technologies. These innovations have the potential to revolutionize biofilm research and clinical practices by enabling quicker diagnoses and more personalized treatment options. Herein, we describe some methods that, formerly developed for other branches of science, can also have noticeable applications in the food field, particularly in the study of biofilms.

### 6.1. Optical Methods

Early detection of biofilms is crucial for adequate control. Among the optical methods, Laser Confocal Scanning Microscopy (LCSM) allows the visualization of biofilm structure using fluorescent dyes [95]. On the other hand, scanning electron microscopy (SEM) is helpful in providing detailed images of biofilm morphology [96], so as to give us a deep understanding of biofilm architecture, the eventual microbial interaction, and the utility of a specific antimicrobial treatment. Atomic Force Microscopy (AFM), which allows the analysis of biofilm morphology, its mechanical characteristics, and internal molecular interactions, even at nanometer resolutions [97], has also proven effective in identifying bacterial growth phases, detecting chemotactic responses, and differentiating between bacterial species based on their movement profiles [98,99].

#### 6.1.1. Laser Confocal Scanning Microscopy

In a confocal microscope, both the lighting and detection systems are aligned to target a single diffraction-limited point within the specimen. This precise point is the only area captured by the detector during a confocal scan. The focal point needs to be systematically scanned across the sample to build a complete image, collecting data at one location at a time. One of the key benefits of confocal microscopy is its ability to perform optical sectioning, enabling the reconstruction of detailed three-dimensional images from high-resolution image stacks. As the name suggests, in a laser scanning confocal microscope (LSCM), the acquisition of images or spectroscopic analyses with the highest spatial resolution is achievable by a remote sensing optical system. In this system, images are formed by focusing a laser on the sample using a confocal microscope and collecting the optical signal as a function of the focal point’s position on the sample. Scanning the sample, the focus position is varied, and the image is reconstructed point-by-point. The underlying idea of confocal microscopy is to focus the signal collected by a microscope’s objective onto a spatial filter. This filter consists of an opaque screen with a small hole comparable to an object’s image at the resolution limit of the objective. The spatial resolution of this composite system is better than that of a single microscope without a spatial filter and is close to the theoretical diffraction limit. The improvement results from only the signal corresponding to the image of the spatial filter’s hole reaching the detector.

The main advantages of LCMS lie primarily in improving vertical resolution, that is, in the direction perpendicular to the focal plane. For this reason, applications that require plane-by-plane three-dimensional reconstruction of images of semi-transparent objects, such as biological samples, extensively use this technique. Another improvement introduced by the spatial filter is blocking unwanted scattered light from the sample before it reaches the detector. In practice, the resolution limit is comparable to the wavelength of the laser, which can be selected based on the application, and is generally between 100 nm and 1 μm. Other factors influencing resolution include the optical component system and the wavelength of the laser radiation in the medium where the object is located [100].

LCSM is an important technique for analyzing biofilms formed by pathogenic or probiotic microorganisms. Thus, it can also be used in the food industry. Its capacity to give high-resolution, three-dimensional images of biofilm structures leads to valuable insights into their formation, architecture, and behavior under different conditions. It provides non-destructive imaging and quantitative data. Furthermore, the technique can provide detailed visualization of a biofilm morphology. This gives us more information about biofilm thickness, density, and spatial organization. Such information becomes essential in the study carried out on pathogens such as *E. coli* O157:H7 and *L. monocytogenes*, whose biofilm, generally, can make them more resistant to the cleaning procedures occurring, for instance, in the food industry [57]. In the case of *L. monocytogenes*, the LCSM helped in the study and identification of the most appropriate strategies to control the cross-contamination of this pathogen on cold-smoked rainbow trout [101]. By applying fluorescent dye during the use of LCSM, we can distinguish the concurrent presence of live and dead microbial cells inside the biofilm and evaluate if and how a specific antibiofilm agent is effective or not [102]. This technique also enables spatiotemporal mapping of biofilm viability and structural changes following antimicrobial exposure. By optically sectioning the biofilm and reconstructing it in three dimensions, LCSM allows researchers to observe how different layers of the biofilm respond to treatment—such as surface cell death versus inner-layer resistance. Moreover, LCSM combined with viability staining offers real-time insights into antimicrobial penetration limits, persistence of resistant subpopulations, and localized matrix disruption, making it a powerful tool for studying the efficacy of antimicrobial strategies in a spatially resolved manner. The real-time observation by LCSM allows us to monitor the formation and maturation of a biofilm, leading to the identification of those critical points for inhibiting its formation and maturation, mainly in the case of pathogens. In a study to assess the effectiveness of double treatment with ultrasound and chlorogenic acid against the biofilm of *S. aureus*, LCSM images revealed that the combined treatment led to a sharp increase and severe damage to the permeability of the cell membrane, causing the release of ATP and nucleic acids and decreasing the exopolysaccharide contents in *S. aureus* biofilm [103]. The system provides a deeper knowledge of the mechanism of chlorogenic acid against *Yersinia enterolitica* [104]. By combining genetics and LSCM, the structural dynamics of *L. monocytogenes* EGD-e sessile growth were characterized in two nutritional environments [with or without a nutrient flow], and the possible role of the *L. monocytogenes* agr system was evaluated during biofilm formation, by tracking the spatiotemporal fluorescence expression of a green fluorescent protein (GFP) reporter system [105]. LCSM allows the characterization of the probiotic biofilms and the identification of new probiotics [106]. It led to the observation of biofilm of the probiotic *E. coli* Nissle 1917 in the gastrointestinal tract, permitting the scientists to hypothesize the positive role of this probiotic and its potential application in functional foods [107]. LCSM was also used to investigate how the concurrent action of *Pediococcus acidilactici*, *L. fermentum*-derived biogenic compounds in conjunction with zingerone were capable of fighting the infection [and biofilm] of *P. aeruginosa* [108]. Moreover, more recently, to assess how the postbiotics metabolites of *L. acidophilus* were able to act against the biofilm of *P. aeruginosa* [109].

#### 6.1.2. Atomic Force Microscopy

Atomic Force Microscopy (AFM) is a powerful imaging and force measurement technique. The technique operates by detecting interactions between a fine cantilever tip and the sample’s surface. Depending on the type of analysis required, the tip can either scan the surface at a close but non-contact distance or make direct contact in contact mode. Unlike traditional optical microscopes that gather and focus on light, AFM does not rely on light to produce images. Instead, it captures high-resolution, real-time surface data by physically sensing the sample. This technique allows for the visualization of surface topographies with remarkable precision, achieving resolutions down to a few tenths of a nanometer. In addition to detailed topographical mapping, AFM stands apart from other microscopy methods by providing insights into various material properties such as stiffness, hardness, friction, elasticity, and the interaction forces between the sample and the cantilever tip. As a result, AFM serves as a powerful tool for investigating the surface characteristics of different microorganisms [110]. The technique permits the analysis of the biofilm morphology, its mechanical characteristics, and molecular interactions occurring inside, even at nanometer resolution. AFM does not require extensive sample preparation and allows for in situ imaging of hydrated biofilms under physiological conditions [111]. These aspects make AFM an essential tool in the study of biofilms in different fields, from microbiology to biomedical research, from the environment to food. AFM operates by scanning the surface of a sample with a sharp probe attached to a cantilever. The movement of the probe across the biofilm causes deflections in the cantilever, caused by the intermolecular forces between the tip and the surface, which are recorded and converted into high-resolution 3D topographical images. The technique allows for both structural imaging and force spectroscopy measurements [112]. By measuring forces between the AFM tip and the biofilm surface, it is possible to have a deeper evaluation of mechanical properties like elasticity and adhesion strength. This information is vital. In fact, by assessing the robustness of biofilms and their resistance to mechanical methods used in the food industry, we could counteract them. In particular, AFM-based force spectroscopy allows for the quantification of viscoelastic behavior and cell-surface or cell–cell adhesion forces at the nanoscale. These biomechanical parameters are essential for understanding how biofilms withstand shear forces or mechanical removal in industrial settings. Moreover, such data contribute to predictive models of microbial colonization by providing insight into the initial attachment strength, maturation dynamics, and potential detachment behavior of different strains on food contact surfaces. This enables the development of more effective cleaning strategies and anti-adhesive surface materials. The use of antimicrobial agents can also be monitored by AFM, which lets us observe the structural changes occurring in the biofilm after the antimicrobial treatment and study and optimize the best strategies to control immature and mature biofilm on food contact surfaces. AFM can also ease the study of precise mechanisms of the adhesion of biofilm to different substrates, including those normally used in the food industry, such as stainless steel and plastic. The use of AFM provided exhaustive visualization of biofilm structures formed by foodborne pathogens, such as *L. monocytogenes* and *Salmonella* spp. It allowed us to understand the architecture and development of their biofilms on food processing surfaces, which is crucial for designing effective cleaning protocols. Enhancing AFM-based surface analysis of food-related microorganisms requires a deeper understanding of their application across various microbial groups, including foodborne pathogens, spoilage organisms, and beneficial bacteria. AFM serves as a valuable tool for measuring size variations and surface characteristics of pathogenic and spoilage microbes, helping to uncover bactericidal mechanisms and cellular adaptations in challenging environments. By examining the adhesion properties of foodborne pathogens such as *L. monocytogenes*, *S. enterica*, and *Bacillus subtilis*, AFM contributed to biofilm management in food-processing settings, reducing the risk of disease transmission and extending food product shelf life [113,114,115,116]. AFM studies on *P. aeruginosa* biofilms demonstrated how EPS components influence biofilm mechanical stability and resistance to antibiotics [117]. The use of AFM allowed us to understand the role of those forces involved in the *Staphylococcus* adhesion to different types of biomaterials [118]. Recently, the concurrent use of AFM and machine learning-based data analysis improved the biofilm characterization techniques [7]. Beyond pathogens and spoilage microbes, AFM is increasingly used to study probiotic bacteria, such as the *Lactobacillus* species, to explore their adhesion properties on both inert and biological surfaces, such as milk [119]. AFM was helpful in the evaluation of the mycotoxin detoxification from beverages using biofilms of LAB [120,121]. The use of AFM contributed to deeper knowledge about the role of *Lactobacillus* biofilm in inhibiting the pathogenic biofilm formation in the food field [122]. The application of AFM for multiparametric analysis of microbial cell surfaces continues to expand, offering new perspectives on the relationship between microbial structure and function. AFM was employed to analyze the basis of the nanomechanical changes occurring in two *Lactobacillus* strains under nitrofurantoin, furazone, and nitrofurazone exposure, recording significant changes in the two strains’ cell morphology, topography, and adhesion parameters [123]. Recent research, also with the use of AFM, highlighted innovative approaches to biofilm control, including the use of probiotic *Lactobacillus* as an alternative to traditional antibiotics, offering promising potential for future biofilm treatment strategies [122,124].

#### 6.1.3. Scanning Electron Microscopy

SEM gives three-dimensional images of the biofilm structure and surface morphology. Through SEM, we can investigate microbial adhesion, the thickness of a biofilm, and the composition of EPS, even with nanometer resolution [125]. SEM has become a fundamental tool in biofilm studies across various disciplines, including medical microbiology, environmental science, and industrial biofouling research, but could also be highly appealing for the study of biofilm in food environments. SEM generates high-resolution images by scanning a sample with a focused beam of electrons. The interaction occurring between the electron beam and the sample surface produces various signals, such as secondary electrons [SE] and backscattered electrons (BSE). They are collected and form detailed images of biofilm structures [126]. To visualize biofilm by SEM, the samples are fixed to maintain their structure; then, the samples are dehydrated, and the liquid is replaced with a gas to reach the so-called critical point. To prevent electron charging, a thin layer of conductive material, such as gold, carbon, or platinum, is applied. At this point, the coated biofilm can be exposed to an electron beam, which gives rise to signals that are captured to produce images. Generally, the types of SEM signals that are used in the study of biofilm are secondary electrons (SEs), which give high-resolution images of the biofilm surface morphology; the so-called backscattered electrons (BSEs), used to detect the interaction among bacteria or the eventual variation in the EPS content; and the energy-dispersive X-ray spectroscopy (EDS/EDX), through whom we might identify the elemental composition occurring within the biofilm. This signal is used to analyze mineral deposits, antimicrobial coatings, or the metals coupled to biofilms [127]. The use of SEM is of relevance in the study and research on biofilms. It allows us to visualize microbial adhesion, biofilm maturation, and the distribution of EPS, providing more insights into the steps occurring during biofilm formation [128]. Through SEM, we can investigate the surface coverage and density of a biofilm, how the bacteria are arranged within the matrix, and, interestingly, the pore networks and water channels present within a mature biofilm. SEM studies revealed dense EPS networks involved in the antibiotic resistance exhibited by *P. aeruginosa* [129]. SEM allows for the investigation of medical devices [130] and can also be helpful in the study of more appropriate materials to be used in food packaging to detect eventual corrosion or contamination [131]. Such studies could also be useful in the study of complex biofilm structures, for example, those formed by bacteria and yeasts [132]. SEM could obviously be used to compare the integrity of biofilm structures before and after treatment, but can also be used to identify what damage occurs in bacterial cells and if there are some changes in the distribution of the exopolysaccharides following the antimicrobial treatment. For instance, SEM imaging showed cell lysis and EPS disruption in biofilms treated with silver nanoparticles or sodium acid sulfate, confirming their bactericidal effects [133,134]. Through SEM, the growth, survival, and biofilm formation of *S. enterica* in the presence of high and low concentrations of catfish mucus extract on four food-contact surfaces and observed the biofilm populations were investigated [135]. In this case, the surface properties, surface roughness, and surface energies were determined using contact angle measurement and Atomic Force Microscopy. However, despite its advantages, traditional SEM has some limitations when applied to biofilm research: first, the steps of fixation and dehydration can alter the biofilm structure; furthermore, the surface imaging does not give us deep insights into the biofilm architecture. Some biofilms may be sensitive to electron exposure, giving rise to distortions. SEM overlapped such limitations through the Cryo-SEM, which does not use chemicals to fix and dehydrate biofilm but freezes samples in liquid nitrogen before imaging. This allows us to have more accurate representations of biofilm morphology [136]. The Focused Ion Beam Scanning Electron Microscopy [FIB-SEM] enables the three-dimensional (3D) reconstruction of biofilms by precisely cutting thin layers and capturing images of each section. This technique provides a detailed view of the internal biofilm structure, allowing for in-depth analysis of its composition and organization [137]. Finally, through the environmental SEM (ESEM), we have biofilm imaging without the necessity for extensive sample preparation. This allows for the real-time observation of biofilms that, when hydrated, show their natural state [127].

### 6.2. Microfluidics

Microfluidic technology is a multidisciplinary field involving the handling of liquids at the microscopic scale, generally in channels with dimensions ranging from a few micrometers to a few millimeters. These platforms facilitate precise control over various parameters, such as flow, temperature, and chemical gradients, allowing the creation of highly controlled environments that reproduce physiological conditions [138]. The importance of microfluidics in the study of biofilm formation on food surfaces is particularly pronounced because it allows us to study not only the complex dynamics of microbial interactions but also to imitate natural environments in which these biofilms develop. Microfluidic platforms offer the capacity for observation and real-time analysis of the training of biofilms, providing an overview of the stages of development and the maturity of biofilms. Microfluidic platforms have revolutionized the ability to study the formation of biofilms in controlled environments that closely resemble the physical-chemical properties of food surfaces. Research conducted by Pérez-Rodríguez et al. [139] highlights how microfluid devices allow imaging and real-time monitoring of starter biofilms developing on surfaces, such as those encountered in the processing of dairy products. The capability of manipulating environmental variables, such as the composition of nutrients, the flow rate, and temperature within these devices, provides information on the dynamics of the formation and stability of biofilm. This understanding is fundamental as it reveals how specific starters can not only dominate pathogenic species, but also how they behave in variable conditions that simulate food production scenarios of the real world. Microfluidic platforms have become powerful tools to study the formation and characteristics of biofilms by pathogenic bacteria, particularly pathogens of food origin, such as *E. coli* or *L. monocytogenes*, on various food surfaces [138], providing information on the mechanisms of attachment and maturation of biofilms, and revealing the environmental factors affecting such processes, making it possible to identify potential intervention strategies aimed at preventing foods of food. These platforms allow us to create controlled micro-environments that imitate the complex conditions of food production and consumption, thus providing an overview of the architecture and functionality of the biofilm. Tremblay et al. [140] used microfluidic devices to observe the formation of *E. coli* in various conditions of nutrients and flow, providing critical data on conditions that promote biofilm resilience, which are essential for developing effective cleaning protocols in food processing areas [140]. Al Ghamdi et al. [141] expanded these results by incorporating a multi-species approach to their microfluidic model, allowing the examination of interspecies interactions within biofilms. Their research pointed out that pathogenic bacteria can coexist with nonpathogenic strains, improving the overall resilience of biofilms against antimicrobial agents. This interaction complicates food security measures, as traditional efforts to eradicate unique species could inadvertently support the survival of a more robust biofilm community. By adapting these microfluidic platforms for antibiotic sensitivity tests, the therapeutic potential of various antimicrobial agents in real time can be effectively assessed, allowing a more nuanced approach to fighting against biofilms in food environments. The differential conditions established in microfluidic contexts may simulate the scenarios encountered during food processing (such as different levels of pH and the availability of nutrients), thus informing the development of more robust microbial formulations [142]. The implications of the use of microfluidic platforms extend beyond basic research, with significant advantages for the food industry concerning the management and control of biofilm. A detailed analysis of the kinetics and the architecture of biofilms makes it possible to design new cleaning protocols, to assess the effectiveness of antimicrobial agents, and, ultimately, to improve food preservation methodologies. Furthermore, a deeper knowledge of the different degrees of biofilm interactions—including competition, cooperation, and the influence of non-viable components—can provide important information about the development the food preservation strategies. By filling the gap between fundamental microbiological research and practical food security applications, microfluidics can become a transformative approach to improve our understanding of microbial biofilms and their implications for food integrity. Figure 3 shows a microfluidic platform for the evaluation of the effectiveness of an antibiotic in fighting a bacterial biofilm.

By adopting microfluidic platforms to advance our understanding of the training of pathogenic biofilm, the food industry can better alleviate the risks associated with pathogens of food origin, ultimately improving public health and food preservation efforts. The advent of microfluid platforms provided deeper information about the interactions between probiotics and pathogens transmitted by food, revealing the dynamic nature of biofilm formation processes. Microfluidic devices allow precise control of environmental parameters and real-time observations of microbial interactions in a microsite, providing an incomparable opportunity to investigate the dynamics of the biofilm. Microfluidic devices have been used to study the behavior of probiotic and starter cultures, which are also important to maintain the quality, safety, and health properties of food during their manufacturing and storage. Recent studies indicated that probiotics can significantly reduce the adhesion of pathogens to surfaces by altering the physical and chemical properties of the biofilms. For example, Xiang et al. [143] used a microfluid system to systematically analyze the interactions between *L. rhamnosus* and *S. enterica* in a controlled environment, revealing that the presence of *Lactobacillus* effectively inhibited the initial adhesion of *Salmonella* cells, which suggests a mechanism of competitive exclusion. This microfluidic approach not only illuminated the initial stages of biofilm formation but also allowed the evaluation of microbial behavior in variable environmental conditions, such as the availability of nutrients and cutting forces. Similarly, Meroni et al. [144] explored the facilitative role of probiotics in the modulation of biofilms formed by pathogenic fungi on food contact surfaces. When using a microfluid platform, they were able to simulate the environments of the food industry where these interactions usually occur. Their results indicated that probiotics suppressed the growth of pathogenic fungi such as *Aspergillus* spp. and facilitated the establishment of a stable probiotic biofilm matrix, which has improved antifungal properties. This interaction clarifies the broader implications of probiotic biofilms, such as the potential to act as protective barriers against deterioration and pathogenic organisms on food surfaces. The mechanistic understanding of how probiotics inhibit the formation of pathogenic biofilms can also lead to the development of innovative bio-preservation strategies.

By taking advantage of natural antimicrobial properties observed in probiotic strains, researchers can formulate food preservation methods that promote selective growth of beneficial microorganisms while inhibiting the harmful ones. Through microfluidics, we can monitor how the integration of probiotics into packaging materials or coatings could establish a bioprotective layer in food products, which represents a promising route to extend their shelf life by reducing microbial load. Microfluidics highlighted the relevance of a specific architecture in these interactions. The three-dimensional structures formed by probiotics are clearly different from those of pathogens, which can influence how they interact with each other. Understanding spatial disposition and nutrient gradients within these biofilms is fundamental. Advanced image techniques incorporated into microfluidic configurations allow us to visualize and quantify the structural dynamics of biofilms, shedding light on how probiotics can interrupt the formation of pathogenic biofilms through mechanical and chemical means. The double role of probiotics as beneficial microorganisms and competitive pathogens emphasizes their potential as a biological control mechanism in food security protocols.

Microfluidics can also play an important role in the study of the starter cultures, essential in the fermentation processes, based on the production of various dairy and fermented foods, which can significantly influence the organoleptic qualities of food products, while improving their safety and their lives [145,146]. If inoculated on food surfaces, starters can establish a robust biofilm, which is a structured community of microorganisms that adhere to the surfaces and expel a protective extracellular matrix. The formation of a biofilm by the starters is a crucial factor in the competitive movement of pathogenic bacteria, as highlighted by recent progress in microfluidic technologies. Yuan et al. [147] demonstrated the competitive interactions between *Lactobacillus* spp. used in the production of yogurt and the pathogens of food origin, such as *L. monocytogenes* and *E. coli*, underlining the potential of the starters as natural biocontrol agents. Through the formation of biofilm, these bacteria can create a physical barrier that inhibits the adhesion and proliferation of pathogens, thus improving food safety. The microfluidic platforms facilitate the analysis of these interactions on a microscale, allowing researchers to view and quantify the specific conditions that favor the domain of the starters. In addition to their defensive mechanisms, the starters contribute to the conservation of food through the production of various metabolic by-products, including organic acids, ethanol, and antimicrobial peptides during fermentation, which further prevent the growth of deterioration and pathogenic agents. Understanding the metabolic pathways involved in these interactions is essential for optimizing the starter culture formulations. Microfluidic technology is decisive in this aspect, allowing high-performance screening of various microbial strains to identify those that have superior biofilm formation skills and antimicrobial properties [148,149]. The implications of this research are substantial for the dairy and fermented food industries in the context of the demand for safer and healthier food options. By exploiting the knowledge acquired by microfluidic studies, manufacturers can adapt starters to improve fermentation processes, improve the quality of the product, and extend the conservation duration. This is particularly relevant, given the growing interest in “clean label” products, in which consumers are increasingly wary of synthetic preservatives and food additives. Microfluidic platforms provide information on the behavior of probiotics and initial starters, which are essential in food preservation and fermentation processes. The resilience of beneficial biofilms against environmental stress can be quantified, and microfluidic channels can be used to evaluate the adhesion and formation of *L. plantarum* biofilms [150]. Microfluidic platforms are powerful tools to elucidate the underlying mechanisms of resistance and persistence of biofilms, particularly in food environments where they can interact with pathogenic, probiotic, and initial crops. These microscale systems allow complex experimental designs that can replicate the microenvironment of food surfaces, facilitating the observation of biofilm behavior in a controlled environment. Gomes and Mergulhao [151] used microfluidic devices to study the resistance of *P. aeruginosa* to disinfectants, revealing critical supplies in the spatial heterogeneity of biofilm architecture and its impact on susceptibility to antimicrobial agents. The ability to manipulate fluid flow and examine biofilm responses to variable concentrations of real-time disinfectants offers a comprehensive understanding of biofilm dynamics that traditional culture-based methods cannot provide. In addition, Mirghani et al. [152] expanded on these findings through the investigation of survival strategies used by *L. monocytogenes* biofilms in food processing environments, showing that *L. monocytogenes* exhibits a pronounced resilience against typical sanitation practices. The advantage of gradient formations within microfluidic devices allows them to observe the differential gene expression related to the mechanisms for stress in biofilms, which indicates how certain cells within a biofilm population can resist disinfectant exposure, while others may be more susceptible. This heterogeneity not only complicates biofilm control efforts but also underlines the need for strategies that address the diversity of biofilms in food security management. The research progress made in recent years highlights the importance of integrating microfluidic studies into broader food safety strategies, with the potential to improve not only food safety but also the quality and functionality of food products through an improved management of pathogenic and beneficial biofilm formers. Increasingly recognized for their impact on food security and quality, biofilms formed by multimicrobial species pose a considerable challenge in food processing and preservation contexts. Microfluidic platforms thus become a powerful tool to elucidate the complexities of these biofilms by allowing real-time observation of microbial interactions in controlled environments that closely imitate the conditions found on food surfaces. Studies using microfluidic systems reveal that interactions between various microbial species in biofilms can considerably modify their structural integrity, resilience to environmental constraints, and global functional capacities [153]. The dynamics of microbial interactions can lead to synergistic or antagonistic effects, which influence the growth and viability of pathogenic organisms and the adhesion and stability of probiotic and starter cultures [154,155]. Microfluidic studies also showed that the spatial distribution of different species in biofilms is essential for their performance and their function in food environments and can reflect a structured ecological community that is able to survive in unfavorable conditions, presenting important implications for food preservation strategies. Distinct areas in a biofilm of mixed species can facilitate metabolic exchanges that improve the availability of nutrients, thus supporting the viability of less competitive strains [153]. Microfluidic platforms allow the exploration of the influence of environmental factors—such as temperature, flow rates, and nutrients—on the behavior of biofilms with mixed species. These ideas are crucial because they shed light on the development of more effective cleaning and sanitation practices in food production facilities. By understanding how biofilms react to variable external stimuli, food scientists can design intervention strategies that disrupt harmful biofilms while supporting beneficial strategies. Tackling these complexities is imperative for improving biofilm management practices, which is a central concern in the continuous effort to optimize food security and conservation in increasingly complex food systems. Microfluidic simulations also provide valuable mechanistic insights into biofilm heterogeneity, interspecies synergy, and antimicrobial resistance profiles. By mimicking the physiologically relevant conditions of food processing environments—such as flow dynamics, nutrient gradients, and temperature shifts—these platforms reveal how structural stratification within biofilms affects antimicrobial penetration and tolerance. Moreover, they help elucidate synergistic interactions in mixed-species communities, where metabolic cooperation or protective layering enhances overall biofilm resilience. Microfluidic devices allow precise monitoring of real-time responses to antimicrobial agents, capturing differential susceptibility among coexisting strains. This information can be directly used to improve predictive models of surface colonization and resistance, guiding the development of more targeted, adaptive strategies for biofilm control in the food industry. A prominent advance is the application of advanced image techniques within microfluidic systems. Traditional methods for the study of biofilms often involve cumbersome processes that produce a limited temporal and spatial resolution, which finally hinders a comprehensive understanding of microbial interactions within biofilms. Recent innovations, such as high-resolution time periodic microscopy and fluorescence recovery after photobleaching [frap], allow the detailed visualization of biofilm architecture and its changes over time [10,156]. The microfluidic-based platform integrated with images of living cells traced the formation of pathogenic biofilms in real time, allowing us to monitor the initial stages of the establishment of biofilms and to observe the interactions between pathogenic and probiotic strains, offering a deeper vision of competitive inhibition and microbial synergy [143]. The recent progress of microfluidic platforms has considerably deepened our understanding of the formation of biofilms on food surfaces, in relation to pathogenic, probiotic, and start-up cultures. These technologies provide an unprecedented overview of the complex dynamics of microbial interactions, allowing real-time monitoring and analysis of the development of biofilms under controlled environmental conditions. Anjum et al. [157] showed that microfluidic systems can simulate food surfaces and conditions, allowing researchers to study the adhesion, growth, and structural organization of high-resolution and temporal microbial communities. Microfluidics helped the research to explore how the various flow rates and shear forces affect the formation of biofilms by pathogens of food origin, such as *Salmonella* and *L. monocytogenes*, elucidating critical factors that improve their colonization [158], with interesting implications for food security and preservation strategies. Concurrently, improving probiotic biofilms and starting through microfluidic studies could lead to an improvement in fermentation processes and the establishment of beneficial cultures, thus contributing to the quality and safety of food. The ability to handle microscopic microenvironments thus offers fascinating possibilities to optimize conditions that promote desirable microbial activities while inhibiting pathogenic activities.

### 6.3. Biospeckle Laser Technology [BLT] in Microbial Studies

Biospeckle technology is an innovative tool with applications in food quality assessment, food safety, and nutraceutical research. This method enables the rapid detection and monitoring of bacterial presence and growth within both solid and liquid biological matrices. Moreover, by analyzing variations in biospeckle patterns, researchers can effectively evaluate and optimize the storage duration of probiotic bacteria encapsulated in alginate, as well as their survival under simulated gastrointestinal conditions [159,160]. For stationary objects that scatter light, the dispersed light produces stable laser speckle patterns. However, when moving particles, such as those suspended in a fluid, exhibit autonomous motion like Brownian motion, individual speckles appear to fluctuate irregularly in brightness, creating a visual effect often described as “twinkling” or “boiling”. This dynamic behavior is referred to as “time-varying speckle”. The biospeckle technique has been successfully validated for monitoring particle motion in optically complex environments by analyzing these fluctuating laser speckle patterns [161]. Analyzing laser speckle patterns provides a powerful means of assessing activity at both microscopic and macroscopic scales. Correlation-based methods are widely used to evaluate time-dependent variations in laser speckle patterns. When a rough surface undergoes deformation, vibration, or displacement, the corresponding shift in speckle patterns can be detected [162]. The extent of displacement can be determined by identifying the peak shift in the cross-correlation function between sequential frames. In cases where a frame is compared to itself, the autocorrelation peak remains at zero. However, a shift in the cross-correlation peak between frames indicates a relative variation between them. The correlation coefficient serves as a valuable metric for analyzing the activity of a time-varying speckle pattern [163]. By comparing each frame in a sequence with the preceding one, variations in the correlation coefficient over time can be measured. Minimal fluctuations in the coefficient within a given area suggest low activity levels within the detection range. A plot of the correlation coefficient vs. time can effectively describe the activity of the observed process [160,163]. Recent research highlights substantial advancements in biospeckle laser techniques and their potential for investigating dynamic processes in microbiological environments. Laser speckle-based methodologies have been successfully applied to assess bacterial chemotactic responses on agar plates [164] and to distinguish motile bacteria from fungi [165]. Additionally, speckle decorrelation time mapping has demonstrated effectiveness in detecting *E. coli* [165] and *Bacillus cereus* on chicken breast meat [166]. Speckle analysis has been employed to assess biomass growth kinetics in liquid cultures [167,168] and determine bacterial susceptibility to antibiotics [169]. The ability of laser speckle techniques to provide rapid results in minimum inhibitory concentration (MIC) testing has underscored their importance, particularly when combined with deep learning and Artificial Neural Networks [170]. In general, the biospeckle laser method offers a simple yet highly effective approach, enabling faster microbiological activity detection compared to conventional methods based on turbidity measurements or manual colony counting.

From a mathematical standpoint, speckles can be interpreted as a random variation within the complex plane [171]. When a coherent light beam interacts with a rough surface, it produces an interference pattern consisting of alternating bright and dark regions, which can be detected by an imaging sensor. If the surface irregularities are comparable in scale to the wavelength of light, the surface can be modeled as a collection of *L* individual scattering points. These scatterers behave as *L* distinct point sources, each occupying a unique three-dimensional position. As light scatters from each point, it traverses different optical paths through media such as air, phosphate-buffered saline (PBS), or water. The detector then captures the intensity of the coherent sum between all the superimposed wavefronts. The phase of each complex wavefront is sensitive to any tiny variation in the optical path, being proportional to the refractive index of the medium and the physical length traveled towards the sensor. Within the coherent sum, the optical path delay differences are multiplied by a factor inversely proportional to the light wavelength. In the case of visible light, this factor is of the order of 10^7^, and thus it largely amplifies any tiny difference between the phase contributions of each addend of the coherent sum. For this reason, speckle techniques are very sensitive to small displacements in time and space, provided these are at least of the order of the light wavelength. Due to its capability of influencing the light paths over time, from an optical standpoint, a biofilm of live self-propelling bacteria can be studied as a random moving diffuser. Treating a biofilm as a moving diffuser has allowed seeing through its turbidity, a useful capability to investigate the layers hidden underneath [172]. Furthermore, the size, spatial distribution, and occurrence patterns of speckle grains can be linked to the shape of the scattering object, its surface roughness, and the density of scatterers either on its surface or within its volume [173]. Due to its exceptional sensitivity, optical speckle metrology has gained widespread adoption in various industrial sectors for non-invasive testing of large structures [174]. Among techniques that leverage speckle patterns to extract information about biological activity within a given field of view (FoV), two primary categories exist: **static approaches**, where the speckle pattern remains unchanged over time, and **dynamic approaches**, where speckle fluctuations highlight the presence of motile microorganisms within a solid or liquid matrix. These dynamic speckle variations contain valuable data regarding object movement or the mobility of particles within a sample [175]. Biospeckle laser technology (BLT) presents several advantages, making it highly suitable for investigating microbial growth dynamics and behavior. It enables the real-time, wide-field analysis of spatiotemporal growth patterns and the extraction of critical phenotypic data. Ansari et al. [176] utilized BLT to investigate bacterial motility and biofilm formation with a temporal resolution of under one second. Their study uncovered intricate dendritic structures with fractal-like growth patterns and rapid movement on surfaces. One of BLT’s key strengths is its non-contact nature, which permits the continuous observation of microbiological samples over extended periods without interference. Van der Kooij et al. monitored biofilm growth and maturation using uninterrupted BLT measurements, requiring minimal sample preparation compared to traditional methods that rely on external markers or staining [177]. The label-free nature of the method and the absence of invasiveness allowed for prolonged quantitative analysis. Their findings demonstrated BLT’s ability to reveal both individual diversity within microbial populations and the three-dimensional architecture of biofilm clusters by analyzing temporal speckle variations [177,178]. Radial expansion rates measured through BLT have shown strong agreement with growth curves obtained using conventional techniques such as optical density measurements and colony counting. BLT was used to track the radial expansion of *Staphylococcus aureus* colonies on agar. The radial expansion curves obtained via BLT closely matched OD600 growth curves, further validating BLT’s reliability for quantitative growth analysis. BLT enables continuous, long-term time-lapse imaging to analyze microbial growth patterns influenced by genetic factors and environmental conditions [179]. Additionally, BLT allows precise morphological analysis, including fractal-like branching patterns and filamentous fungal differentiation [159]. Al Ogaidi et al. reported the use of BLT to characterize the filamentous structure and growth dynamics of *Neurospora crassa*, offering insights into hyphal movement and branching network formation [178]. Dynamic speckle pattern variations further reveal species-specific growth signatures [180]. Ansari et al. demonstrated that BLT can differentiate between *E. coli*, *S. aureus*, and *Klebsiella pneumoniae* by analyzing distinct temporal and spatial texture patterns. Specifically, a hemisphere-capped cylinder model was used to estimate the length and diameter of individual *E. coli* bacteria [176,177,178]. Fourier transform light scattering analysis has also facilitated the classification of four bacterial species based on their rod-like structures [181]. Developing rapid, highly sensitive, and specific detection methods for foodborne pathogens is essential for food safety. Such systems are crucial for preventing the spread of harmful bacteria in the food supply, as extensively discussed by Liu and Ngadi [182]. The simplicity of the experimental setup, requiring only coherent light illumination, makes it a practical tool for microbial studies [160,176,183]. Dynamic speckle techniques have been effectively used to characterize bacterial growth and differentiation. For example, time-lapse BLT imaging distinguished *E. coli* activity at different growth stages, separating infected from non-infected regions. The approach has also been applied to assess bacterial chemotaxis [164] and differentiate between bacterial species based on their distinct activity profiles [184]. Recently, BLT has been coupled to holographic 3D tracking to evaluate the performance of different probiotic strains microencapsulated in alginate. The capability to survive encapsulation and the motion dynamics have been evaluated to rank the probiotic candidate strains to serve as probiotics [185]. An example is provided in Figure 4, which reports (a) a typical biospeckle apparatus, (b) recorded speckle patterns, and biospeckle decorrelation curves obtained under simulated gastrointestinal conditions (c,d) before and after post-processing, from which the candidate can be ranked.

Furthermore, BLT has proven superior to conventional methods in distinguishing motile bacteria from filamentous fungi on agar plates [165,186]. By analyzing speckle variations, researchers have identified bacterial growth phases, such as exponential proliferation and stationary-phase sporulation or cell lysis [187]. BLT has been successfully employed to detect *E. coli* and *Bacillus cereus*, facilitating differentiation between fresh and contaminated water [166,183]. Recent advancements in machine learning have further expanded the capabilities of AI-assisted biospeckle techniques. Multiple descriptors now allow for the categorization of distinct biological activities occurring across different specimen areas [188]. We foresee a higher interconnection between AI and coherent imaging. For instance, advances in hyperspectral imaging are a unique opportunity to characterize the presence of different coexisting strains. It poses several questions, though, since a hypercube is not trivial to analyze. In this framework, the use of machine learning and, above all, deep learning methods, could leverage hyperspectral imaging to exploit its full potential [182,189,190,191,192,193]. This method has even been used to digitally distinguish healthy from bruised apple regions [194]. BLT has also been utilized to analyze the growth patterns of *Bacillus subtilis* strains, comparing wild-type colonies with genetically modified mutants lacking motility and extracellular matrix production genes. This approach provided valuable insights into the mechanobiology of microbial colonies [160]. Long-term BLT studies have also quantified bacterial radial growth rates, demonstrating a strong correlation with conventional optical density and plate counting methods for both Gram-positive and Gram-negative species [195,196]. Additionally, BLT has facilitated the study of bacterial swarming behavior and fractal-like branching structures, offering insights into microbial developmental adaptations [176,197]. In quorum sensing studies, BLT has been employed to track bacterial motility at varying concentrations, linking Autoinducer-2 signaling to both increased activity and local alignment movements. This highlights BLT’s ability to investigate microbial communication mechanisms [197]. Moreover, such technology has enabled the automated monitoring of bacterial growth characteristics across multiple isolates by measuring temporal speckle fluctuations, providing rapid phenotypic diversity assessments [161]. BLT has also been applied to high-throughput antibiotic susceptibility testing, where changes in speckle fluctuations were used to evaluate the dose-dependent bactericidal effects of various drugs. This method allowed for the precise determination of minimum inhibitory concentration [MIC] values, demonstrating its potential for automated antimicrobial screening and precision medicine applications [170]. Future advancements could further enhance the efficiency of BLT-based screening, enabling it to process even larger datasets and making it more suitable for industrial and clinical applications. The integration of microfluidic technology and automated sampling systems could facilitate continuous real-time analysis.

### 6.4. DNA-Based Methods [qPCR, NGS]

Among the continuation of methods based on DNA developed for microbial analysis, the chain reaction of quantitative polymerase chain reaction (QPCR) and next-generation sequencing (NGS) have become powerful tools for the identification of microbial species and genetic markers in biofilms. The exploration of biofilms by methods based on DNA, such as QPCR and NGS, highlights their immense significance not only in microbial ecology but also in biotechnological innovations. While these molecular tools continue to progress, they promise to provide more in-depth information on complex relationships within biofilms, paving the way for enlightened health, environmental remedies, and bioengineering strategies. The integration of these methods will undoubtedly lead to a more holistic understanding of the dynamics of microbial communities and their functional roles in various ecosystems.

#### 6.4.1. QPCR

QPCR is a very sensitive technique that allows the quantification of specific DNA sequences, allowing researchers to assess the abundance of target microorganisms in the biofilm matrix. This capacity is essential in microbial ecology, where understanding the dynamics of the population and interactions within biofilms can elucidate key ecological functions. Recent developments in QPCR tests, including probes and multiplexing, have improved the specificity and flow of microbial identification, facilitating complete analysis of biofilm communities [198]. QPCR has become a pivotal technique for the identification and quantification of microbial species in biofilms, a complex and dynamic community of microorganisms integrated in an auto-product extracellular matrix. The sensitivity of the QPCR allows the detection of microorganisms with low abundance, which are often neglected using traditional methods based on culture. This capacity is particularly important in biofilm studies, where microorganisms can reside in complex networks that contribute to their survival and resilience. Díez López et al. [199] have demonstrated the usefulness of QPCR in the identification of specific bacterial species in dental biofilms, indicating its applicability in oral microbiology, an area where understanding microbial diversity is crucial to manage oral health. In addition, Lazarevic et al. [200] used QPCR to quantify the genetic markers associated with antibiotic resistance in biofilms formed in clinical areas, further emphasizing the relevance of the method in public health and clinical microbiology. The specificity of the QPCR is another crucial aspect that improves its usefulness in microbial ecology. By designing primers that target the unique sequences of the 16S RNA gene or other functional genes, researchers can differentiate with precision between closely related microbial species, thus offering an overview of the dynamics of microbial communities and interactions in biofilms [199]. This precision contributes to elucidating the roles of specific taxa in the formation, stability, and pathogenicity of biofilms. In addition, the QPCR can be adapted to multiplex tests, allowing simultaneous detection of several targets in a single reaction, which is particularly advantageous in complex biofilm matrices.

Further than the identification of pathogens, QPCR has proven promising in monitoring the expression of virulence factors that can be crucial for their pathogenicity. Some studies have demonstrated the effectiveness of the use of QPCR to monitor the genetic expression of biofilm genes in *S. aureus*. This surveillance has enlightened important aspects of its pathogenicity and its resilience in food systems, providing essential data that can clarify risk assessments and hygiene protocols [200,201].

QPCR also allows the evaluation of microbial diversity in biofilms, revealing the interactions between different microbial species. These interactions can contribute to the stability and persistence of biofilms in food environments, ultimately affecting the dynamics and safety of deterioration. By profiling various microbial populations in biofilms, researchers can identify the main bacterial taxa that play a role in deterioration and develop targeted interventions to mitigate the risks associated with food agents associated with biofilm. In terms of deterioration prevention techniques, ideas taken from QPCR analyses can shed light on the development of effective cleaning and disinfecting strategies. By identifying specific genes that contribute to the training and resilience of biofilm, food processors can design targeted treatments that disrupt the development of biofilm at the critical points of the food supply chain. This knowledge is particularly relevant given the growing demand of consumers for mini-transformed foods, which may not undergo in-depth preservation treatments, which generally inhibit microbial growth. Overall, the application of QPCR in understanding biofilm formation in food systems has a multidisciplinary approach that combines molecular microbiology, food safety protocols, and deterioration prevention strategies. The ability to quickly and accurately assess the genetic factors associated with biofilms and their constituents improves the understanding of microbial threats to food safety and lays the basis of innovative approaches to ensure the quality and food integrity in the face of the continuous challenges of food production and distribution. In addition to food security, QPCR plays a crucial role in understanding microbial diversity in biofilms present in food systems. The various microbial communities that form biofilms can vary considerably depending on environmental factors and food matrices. Using QPCR to analyze the composition of microorganisms associated with biofilm, researchers have identified various species, in particular, pathogens of food origin and deterioration organizations, highlighting complex interactions within these communities [202,203]. This molecular technique can discern the relative abundance of different microbial taxa in biofilms, elucidating the structure and dynamics of biofilm consortia, thus highlighting the way these communities adapt to specific environmental conditions. Recent studies have used QPCR to follow changes in microbial diversity in biofilms on surfaces in food processing environments, demonstrating the applicability of the technique in understanding the development and persistence of biofilm [204]. For example, the mapping of changes in dominant species over time has enabled researchers to correlate specific microbial profiles with environmental fluctuations, which can thus help predict the risks associated with foods. These ideas are particularly relevant for food manufacturers because they seek to identify and mitigate potential sources of contamination. The identification of the different microorganisms within biofilms via QPCR extends beyond the simple understanding of microbial diversity. It can identify specific strains which may have increased resistance to cleaning and sanitation protocols, which can compromise food safety; it facilitates risk assessments, and can also shed light on the development of targeted interventions in food security management practices indeed. QPCR application has expanded the understanding of microbial interactions in biofilms. Various studies, such as that of Zhang et al. [205], indicated that co-cultures in biofilms can lead to synergistic or antagonistic relationships that influence global stability and functionality of microbial communities. Understanding such interactions is crucial to designing effective deterioration prevention techniques. By identifying the main players in the development of biofilms, targeted strategies can be implemented to disrupt microbial communication channels and minimize the formation of harmful biofilms, which affect the quality of food. QPCR can be used to study the expression of genes of various inhibitors and regulators of the development of biofilm, thus informing the design of new antimicrobial strategies. Recent studies have highlighted the role of naturally derived antimicrobial compounds in the attenuation of the formation of biofilms. For example, Kim et al. [206] and Wen et al. [207] demonstrated the effectiveness of quercetin and other metabolites against notable pathogens of food origin, such as *S. enterica* and *S. aureus*. These flavonoids can exhibit an impact on the expression of essential genes linked to biofilm, as evidenced by QPCR analyses. Such studies highlight the potential to take advantage of phytochemicals as biocontrol agents to harm the establishment and maintenance of biofilm in food environments, ultimately improving food safety. The QPCR helps to elucidate the signaling routes that pathogens exploit to initiate and develop biofilms. By analyzing the expression of the genes involved in the detection of the quorum, the adhesion, and the formation of an extracellular matrix, the researchers have an overview of the molecular mechanisms facilitating the development of the biofilm. For example, genes like Luxs, which are an integral part of cell–cell communication in bacteria forming biofilms, can be studied through QPCR to determine their expression profiles in different environmental conditions or in the presence of antimicrobial agents. Such information is vital to identify potential intervention targets and to understand how the environmental factors and interactions of the food matrix influence the dynamics of biofilm in real contexts [208]. The implications of these results extend to the development of interventions adapted to specific deterioration organizations, in particular food systems. As QPCR provides a sensitive and specific method to monitor the expression of biofilm genes, food producers can adapt their strategies according to the empirical evidence of the formation and resistance of biofilm. For instance, identifying the overexpression of biofilm synthetic genes in response to certain environmental stimuli can lead to targeted interventions, such as modification of formulations or treatment conditions to minimize the development of biofilm. The ability to monitor microbial diversity within biofilms via QPCR can shed light on the management of deterioration and food security protocols. By discriminating between different microbial species according to their unique genetic signatures, food manufacturers can better understand complex interactions in mixed-species biofilms, in particular, in food processing environments where various microbial communities are spread. This knowledge facilitates the identification of dominant deterioration organizations and informs the design of more effective cleaning and sanitation diagrams. The versatility of the QPCR methodologies facilitates their application through a variety of food matrices, an essential feature considering the different nature of food products and their associated microbial communities. Future research should give priority to the improvement of QPCR techniques to analyze biofilm in various environmental conditions, including temperature, pH, and the presence of competing microorganisms. In addition, the adaptability of QPCR to different types of foods—which move from dairy to meat products—will improve its usability and effectiveness for monitoring food safety [209]. Therefore, it is essential to ensure that such methodologies remain relevant and applicable in the face of the practices and regulations of evolving food transformation. However, the QPCR application is not without challenges. A significant limitation is the bias of the primer, which can result from variations in the affinity of primers for different models, which leads to an over-representation or sub-representation of certain species in the final quantification. Gong and El-Omar [210] stressed that the effectiveness of primer sets can be influenced by the specific environmental context, requiring careful design and validation of primers to ensure precise amplification in various microbial communities. In addition, the optimization of reaction conditions is essential to mitigate the differences associated with various types of samples, such as those derived from various aquatic, land, or clinical biofilms. The optimization of sample extraction methods is essential because the quality and quantity of DNA extracted can influence the reliability of the QPCR results. Variations in the structure of biofilm, thickness, and presence of microbial extracellular polymer substances can further complicate DNA extraction and, therefore, subsequent QPCR results. Consequently, methodological normalization and rigorous validation procedures are imperative to obtain reproducible results in different biofilm environments. Despite these challenges, the advantages of QPCR, including its high sensitivity and specific detection capacities, make it an essential tool in microbial ecology and biotechnology. While researchers continue to refine and adapt QPCR methodologies, this technique will undoubtedly play a central role in improving our understanding of microbial ecosystems and their applications in biotechnological innovations. The complex architecture of biofilms improves microbial resistance to antimicrobials and traditional cleaning methods, which makes pathogens associated with biofilm particularly difficult to control. Using QPCR, the presence of genes associated with the training of biofilms was identified, allowing a precise assessment of potential risks in food production and processing environments. This ability to quantify pathogens forming biofilms provides critical information on relations between microbial diversity, robustness of biofilm, and food security.

#### 6.4.2. NGS Technology

Understanding the DNA sequence is essential for grasping how genes and other genomic elements influence the characteristics of biofilms. Traditional techniques for functional genome analysis include Maxam–Gilbert sequencing, Sanger sequencing, and Shotgun sequencing. However, these older methods have several drawbacks, such as being time-intensive, having challenges with sequence assembly, and offering only short read lengths [211]. In recent years, next-generation sequencing (NGS) technologies have emerged, allowing us to have the complete DNA sequence of a microbial genome in a single run and in a single day [212]. From these data, valuable insights can be gained—not only for strain typing but also for identifying genes related to resistance and virulence, which are crucial for investigating disease outbreaks. NGS has brought about a significant transformation in the field of biology, enabling laboratories to conduct a diverse range of applications and investigate biological systems at an unprecedented level of detail. Contemporary inquiries in genomics necessitate a level of intricacy in data that surpasses the limitations of conventional methodologies of DNA sequencing. The NGS has effectively addressed this gap and has emerged as a ubiquitous tool for addressing these inquiries. NGS technology shares similarities with the Sanger-based capillary electrophoresis sequencing approach. The process of DNA synthesis involves the catalytic activity of DNA polymerase, which facilitates the integration of deoxyribonucleotide triphosphates (dNTPs) that are fluorescently labeled into a preexisting DNA template strand. This incorporation occurs in a stepwise manner through a series of sequential cycles. At the juncture of integration within each cycle, the nucleotides are detected through excitation of fluorophores. The main difference between Sanger sequencing and NGS is that NGS uses highly parallelized techniques to stretch the sequencing of a single DNA fragment to millions of fragments.

Illumina is among the most widely adopted platforms for next-generation sequencing [NGS]. Its workflow is generally divided into four key steps:**Library Preparation**: This step involves randomly breaking the DNA or cDNA sample into smaller fragments, after which adapter sequences are attached to both ends (5′ and 3′) through a ligation process. An optimized method called *tagmentation* streamlines this process by combining fragmentation and adapter ligation into a single step, significantly enhancing overall efficiency. The adapter-tagged fragments are then amplified using PCR and cleaned up—often by gel-based methods.**Cluster Generation**: The prepared library is loaded onto a flow cell containing surface-bound oligonucleotides complementary to the adapter sequences. Each DNA fragment binds to the surface and undergoes *bridge amplification*, a method that produces dense, clonal clusters of identical DNA strands. Once cluster formation is complete, these DNA templates are ready for sequencing.**Sequencing**: Illumina’s sequencing-by-synthesis (SBS) technology utilizes a unique approach involving reversible terminator-bound nucleotides. All four fluorescently labeled dNTPs are added in each sequencing cycle. This concurrent incorporation helps minimize bias and dramatically lowers raw error rates compared to other sequencing platforms, thanks to the natural competition among bases. The approach delivers highly accurate, base-by-base reads and effectively mitigates sequence-specific errors—especially those found in repetitive regions or homopolymer stretches.**Data Analysis**: After sequencing, the resulting reads are mapped against a reference genome. This alignment enables various types of downstream analyses, such as identifying insertions and deletions [indels], detecting single-nucleotide polymorphisms (SNPs), quantifying gene expression in RNA-seq experiments, and performing metagenomic or phylogenetic studies.

The PacBio RS system, developed by Pacific Biosciences, is currently the leading technology in this field. It employs the single-molecule, real-time (SMRT) DNA sequencing technique. The SMRT sequencing methodology relies on the sequencing-by-synthesis approach. An SMRT chip is equipped with numerous zero-mode waveguides, which are utilized to attach DNA polymerase molecules for the synthesis of DNA fragments of interest. In contrast to second-generation sequencing, the most recent SMRT technology can attain an average read length ranging from 5500 to 8500 base pairs [213].

The exploration of pathogens within biofilms has transitioned from relying on traditional, culture-based lab techniques to using molecular tools, and more recently, to modern, culture-independent methods focused on microbiome analysis, such as NGS [214]. Since microorganisms in biofilms are often difficult to grow in laboratory conditions, studying them with conventional clinical methods is particularly challenging. Thanks to NGS and its ability to analyze DNA without the need for cultivation, researchers and healthcare professionals have gained access to microbial communities that were once impossible to examine. This has significantly changed our perspective on microbial diversity [215]. NGS allows for detailed genetic analysis either in a single sequencing run or on a massive scale, handling thousands to millions of genomes each year. Moreover, the decreasing cost of NGS technologies has made it easier to sequence and reconstruct genomes from a wide range of microbial strains, many of which are now stored in public databases like the NCBI.

Microbiome sequence workflow.

To begin the process, biofilm samples are collected, and DNA is extracted using specialized DNA extraction kits. The extracted DNA is then broken into smaller fragments, and specific adapter sequences are attached to both ends to construct the sequencing library. Next, the DNA library undergoes clonal amplification on a flow cell, followed by sequencing using various next-generation sequencing [NGS] platforms. Once sequencing is complete, the resulting reads are aligned to a reference genome, and the data are processed and visualized using dedicated bioinformatics software [215].

Today, the NGS is being used widely for biofilm studies [216,217,218,219]. Shin et al. described the application of NGS technologies in unraveling the complex microbial communities associated with endodontic infections. Their study primarily revolves around analyzing bacteria present in root canal infections, employing advanced sequencing techniques to better understand these intricate microbial ecosystems. Through the use of bioinformatics tools, Shin and al. identified and classified the bacteria present in the samples, finding that the intracanal samples contained the highest concentrations of *Firmicutes*, *Actinobacteria*, *Bacteroidetes*, *Proteobacteria*, and *Fusobacteria* species, and the members of genera *Prevotella*, *Fusobacterium*, *Parvimonas*, *Lactobacillus*, *Streptococcus*, and *Porphyromonas* were strongly related to intracanal samples [220]. In recent years, NGS technologies have allowed the study of pathogens and their biofilm identification [221]. In comparison to other approaches, the results from NGS provide a significant advantage in situations where microbial identification may be impacted by antibiotic use in the past [222]. In food microbiology, NGS can allow the study of complex biofilms, optimal niches for residing microbes that, due to the production of specific metabolites, can lead to alterations in the biochemical characteristics and nutritional and safety properties of the fermented food. In such cases, on the contrary, biofilms can represent the best favorable condition for the production of amino acids like L-proline and L-threonine [223]. In fermented foods, the biofilm sessile cells of *Lactobacillus plantarum* are capable of higher malolactic fermentation than the corresponding planktonic forms of this bacterium [67]. Similarly, the sessile cells of some LABs, such as *L. casei* ATCC 334, can exhibit an improved survival ability under adverse conditions like a low pH environment during malolactic fermentation [224]. NGS thus represents a powerful tool with significant potential for advancing future biofilm research. As these sequencing technologies continue to evolve, their precision will increase, allowing for more detailed analysis of microbial communities within biofilms, down to the species and even strain level. Integrating NGS with other omics techniques—such as long-read sequencing and single-cell analysis—will further enhance our ability to study biofilms. These combined approaches will provide a deeper understanding of biofilm ecosystems, including microbial gene activity, metabolic functions, and adaptation strategies. Through NGS, scientists will be better equipped to explore the complexity of biofilms and develop innovative solutions to address biofilm-associated challenges.

### 6.5. CRISPR-Cas Systems

The advent of CRISPR technology (regularly grouped short palindromic repetitions) transformed the scenario of genetic engineering, offering an unprecedented level of precision in manipulating genetic material. Initially discovered as an adaptive immune system in bacteria, CRISPR emerged as a revolutionary tool for genetic modification in various organisms due to its ability to edit genes with remarkable accuracy and efficiency. The Crispr-Cas system was first discovered in the late 1980s as a sequence of unusual repetitive DNA within the genomes of bacteria, which led to the realization that these sequences played a vital role in the adaptive immune responses of the prokaryotes [225]. The acronym Crispr is often combined with a set of associated proteins known as proteins associated with Crispr (CAS). CRISPR-Cas systems, originally discovered as an adaptive immune mechanism in bacteria and archaea, have been repurposed as highly versatile gene-editing tools. The system is present in roughly 50% of bacterial genomes and 87% of archaeal genomes and serves as an adaptive immune defense mechanism in bacteria [226,227]. The implications of the CRISPR-CAS systems extend beyond their role in microbial immune defense. They have a deep relevance in the context of pathogenicity and resistance to antibiotics. The ability of such systems to direct and split strange nucleic acids, including plasmids and transposons, suggests a natural mechanism to influence the transfer of horizontal genes, an important factor in the dissemination of antibiotic resistance genes between bacterial populations [228]. As pathogenic bacteria acquire resistance features, their CRISPR-CAS systems could play a double role, where they not only confer immunity to the predation driven by phage, but they can also inadvertently improve stability and spread of resistance factors through selective pressure. This raises critical questions regarding the interactions between the CRISPR-CAS systems and commonly used antibiotics, since the abuse or excessive use of these medications could select bacteria capable of evading therapeutic interventions.

The CRIPSR systems function by utilizing sequence-specific RNA guides to direct nucleases toward specific genetic targets, allowing for precise modifications within bacterial genomes. This characteristic makes CRISPR technology a highly promising strategy for combating persistent bacterial biofilms, which are notoriously resistant to conventional antimicrobial treatments [229].

CRISPR-based biofilm disruption strategies aim to selectively target, weaken, and dismantle biofilm structures by interfering with essential biofilm-related genes, degrading EPS, or preventing bacterial communication systems, such as quorum sensing. These interventions ultimately reduce bacterial adhesion and cohesion, making biofilms more susceptible to conventional treatments.

Two of the most promising CRISPR-associated systems for biofilm eradication are CRISPR-Cas9 and CRISPR-Cas3, both of which operate through distinct mechanisms. CRISPR-Cas9 functions as a precision genome-editing tool by inducing double-strand breaks (DSBs) in target genes that are critical for biofilm formation and maintenance. The targeted genetic disruption prevents bacteria from synthesizing key biofilm components, effectively halting their ability to form or sustain biofilm structures [230]. Unlike Cas9, which generates localized gene deletions, Cas3 acts as a processive nuclease with exonuclease activity, meaning it systematically degrades large sections of bacterial DNA. This allows for the complete elimination of biofilm-forming bacteria rather than just impairing their ability to form biofilms [231]. The specificity and adaptability of CRISPR make it an ideal tool for biofilm eradication, as it allows researchers to selectively disrupt biofilm-related genes without affecting beneficial microbes. By engineering CRISPR constructs to target species-specific sequences, it is possible to selectively eliminate harmful pathogens while preserving beneficial microbial communities in environments such as the human microbiome, industrial bioreactors, and water treatment systems.

Several CRISPR-based approaches have been developed to effectively dismantle biofilms. These strategies are designed to either directly eliminate biofilm-forming bacteria or weaken the structural integrity of biofilms, making them more susceptible to additional antimicrobial treatments.

CRISPR-Cas9 and CRISPR interference (CRISPRi) have been employed to silence or knock out critical genes involved in biofilm development, including the following:(1)EPS Synthesis Genes: Genes such as pel, pga, and bcs are essential to produce the extracellular polymeric substances that form the biofilm matrix in *P. aeruginosa* and *E. coli*. Disrupting these genes prevents bacteria from developing a protective biofilm structure.(2)Adhesion and Fimbriae Genes: Genes encoding cell surface proteins [csgA, fimH, and fnbA], which can facilitate bacterial attachment and biofilm initiation in *S. aureus* and *E. coli*, can be selectively silenced using CRISPR to reduce biofilm formation at its earliest stages.(3)Quorum Sensing Regulators: Some genes, such as luxS, lasR, and rhlR in *P. aeruginosa*, are responsible for bacterial cell-to-cell communication. The disruption of such pathways impedes bacteria from coordinating the formation of biofilms, thereby inhibiting maturation and persistence [232].

Thus, by acting against these specific biofilm-associated genes, CRISPR effectively impairs bacterial colonization and weakens the protective EPS matrix, making the remaining bacteria more susceptible to antimicrobial agents and immune clearance mechanisms.

A highly efficient method for introducing CRISPR constructs into biofilms involves the use of genetically modified bacteriophages. Bacteriophage-based CRISPR delivery systems have been developed to selectively eliminate biofilm-forming bacteria while concurrently preserving non-target microbial species. These phages are programmed to deliver CRISPR-Cas nucleases that selectively degrade bacterial DNA, eliminating biofilm-forming bacteria [233]; the system can modify antibiotic-resistant genes to restore the susceptibility of biofilm-embedded bacteria to conventional antimicrobial treatments. By preventing bacterial communication and coordination of biofilm formation, quorum-sensing molecules are disrupted. CRISPR-Cas3 delivered via engineered bacteriophages has been successfully used to degrade entire bacterial genomes, effectively removing *P. aeruginosa* biofilms [234]. This approach has been particularly promising in treating chronic infections, such as cystic fibrosis-associated lung infections and biofilm-associated medical device infections; thus, it could also be used to fight the *P. aeruginosa* biofilm in the food system. Another advanced approach utilizes CRISPR to modify bacterial populations, enabling them to actively break down biofilms. This technique is also realized on probiotic bacteria, which, when modified with CRISPR-Cas, are capable of infiltrating existing biofilms and selectively eliminating pathogenic members. Through the mechanisms of horizontal gene transfer, the system allows us to introduce specific biofilm-disrupting mutations into biofilm-forming bacterial communities, so as to effectively reduce their persistence and resistance to treatments [235]. This approach mimics natural microbial competition and has potential applications in environmental remediation, industrial biofilm control, and human microbiome engineering.

CRISPR-based methods offer several key advantages over traditional biofilm control strategies: they target biofilm-forming bacteria with minimal effects on beneficial microbes, and they have a so-called unparalleled specificity. In addition, unlike antibiotics, CRISPR disrupts essential genetic pathways, making resistance evolution significantly more difficult.

CRISPR is highly flexible, as it can be customized to target different bacterial species due to its programmable and adaptable nature. This strategy can be combined with antibiotics, antimicrobial peptides, and disinfectants to enhance their effectiveness [226]. However, despite its enormous potential, CRISPR-based biofilm disruption faces several technical and practical challenges: first, we should be sure that the CRISPR constructs penetrate all layers of biofilms, which remains a significant hurdle. The human immune system may neutralize CRISPR-based components, reducing their therapeutic efficacy. Future research is focused on improving CRISPR delivery systems using nanoparticles, self-replicating plasmids, and phage-based vectors to enhance targeting precision and efficiency. Additionally, CRISPR’s potential applications in industrial biofilm control and food environmental decontamination are actively being explored [236,237]. CRISPR has been used as a rapid screening method for antimicrobial probiotics [238]. In this case, the scientists used the so-called CRISPRzyme assay, based on a CRISPR-DNAzyme cascade, where the target gene sequentially activated Cas12a protein and DNAzyme, yielding a limit of detection of 62 CFU *Vibrio parahaemolyticus*, 86 CFU *S. Typhimurium*, and 82 CFU *L. monocytogenes*. The elimination of nucleic acid amplification shortened processing time and operational complexity. CRISPR-based technologies have reformed the capacity to manipulate genomes. For example, it has been applied for editing the genome of food-grade lactobacilli and developing therapeutics probiotics [239]. CRISPR-based technologies are also of great potential to alter the genetic content of food bacteria to control the composition and activity of microbial populations across the food supply chain, from the farm to consumer products. Advancing the food supply chain is of great societal importance as it involves optimizing fermentation processes to enhance the taste and sensory properties of food products, as well as improving food quality and safety by controlling spoilage bacteria and pathogens [240].

### 6.6. Organoids

Organoids are 3D clusters of cells grown in vitro that mimic the structure and function of real organs, hence the nickname “mini-organs”. They can be derived from embryonic stem cells (ESCs), induced pluripotent stem cells (iPSCs), or adult stem cells (ASCs), and self-organize into complex tissues through cell sorting and signaling pathways influenced by the culture environment. ASC-derived organoids come directly from adult tissues and grow in media enriched with growth factors that simulate natural conditions. These models are useful for studying normal tissue function and disease and for personalized medicine. ESC- and iPSC-derived organoids, on the other hand, follow staged differentiation protocols to model organ development from all three germ layers. This is especially useful when studying early human development, where access to primary tissue is limited. Three-dimensional culture techniques have advanced from simple cell aggregates to sophisticated systems using scaffolds (like Matrigel) or scaffold-free methods such as hanging drops or air-liquid interfaces. These environments better replicate in vivo conditions compared to traditional 2D cultures. Historically, the concept of cells’ self-organization was demonstrated as early as 1907. Major advances followed, including the isolation of mice and human pluripotent stem cells and the development of iPSCs. In 2009, a pivotal study showed that single adult stem cells could form intestinal organoids in 3D culture without support from surrounding tissue, laying the foundation for organoid research across multiple organ systems—including the brain, liver, lung, and kidney. The intestinal, gastric, and liver organoids are three of the most important ones. Sato et al. first developed long-term 3D cultures of intestinal organoids from single Lgr5+ stem cells in Matrigel, using Wnt agonists, Egf, and BMP inhibitors to recreate crypt-villus architecture [241]. Similar approaches have been used for colon, adenoma, and adenocarcinoma organoids [242]. Human PSC-derived intestinal organoids were also generated through stepwise protocols involving activin A, Wnt3a, and Fgf4 to specify hindgut identity before embedding in Matrigel [243,244]. These organoids contain both epithelial and mesenchymal components and integrate with vasculature upon transplantation [245]. Gastric organoids were first developed from adult mouse pyloric [246]. Concerning liver organoids, adult Lgr5+ cells near bile ducts can form organoids in Matrigel and differentiate into hepatocytes [247].

Based on their complexity, organoid systems are generally grouped into three main types: air-liquid interface (ALI) models, 3D spheroid organoids, and organoid-on-a-chip platforms [248].

ALI models are developed using different cell types cultured on permeable membranes. In these systems, the upper [apical] side of the cells is exposed to air, while the lower (basal) side remains in contact with culture medium. This dual exposure promotes the differentiation and maturation of various specialized cell types, such as those involved in mucus secretion or ciliary motion, resembling native epithelial tissues [249,250]. The compartmental structure of ALI systems, with distinct apical and basal sides, makes them ideal for co-culture experiments to study immune responses to biofilms. Advancements have made ALI models more accessible for high-throughput applications. A miniaturized version using 96-well plates was developed to model human small airway epithelium, enabling broader use in screening studies [251]. A comparable ALI model for skin was created using N/TERT keratinocytes [252], allowing robust biofilm formation from *Pseudomonas aeruginosa*. This model was also used to evaluate the impact of various anti-biofilm peptides on both biofilm disruption and skin barrier integrity. Figure 5 shows a 3D coculture model with a pathogen biofilm.

A more advanced model is represented by 3D spheroid organoids, where progenitor cells are guided through a series of differentiation steps, using specific combinations of growth factors that modulate key signaling pathways [253,254]. These 3D structures closely replicate the natural architecture, multi-lineage cell differentiation, and developmental processes of mammalian epithelial tissues. Their enclosed configuration also creates a unique environment that can support the growth of bacterial species that are typically difficult to culture in standard in vitro systems [255]. Additionally, due to their self-renewing capacity, 3D organoids can be maintained and expanded over extended periods, making them well-suited for the investigation of long-term or chronic infections [256]. By integrating 3D organoids with techniques such as microinjection and live imaging, Forbester et al. [257] demonstrated that Salmonella enterica serovar Typhimurium could penetrate the epithelial barrier and localize within intracellular vacuoles, mirroring in vivo infection patterns. Although 3D organoids and air-liquid interface (ALI) models more accurately emulate physiological conditions than traditional 2D systems, they still lack the mechanical and biochemical complexity of the native environment, including factors like shear stress and nutrient gradients. These limitations can be addressed by merging organoids with microfluidic systems, resulting in organoid-on-a-chip models—platforms that simulate the interaction between host tissues and microorganisms under dynamic, flow-based conditions. By precisely controlling fluid flow to mimic natural intestinal movements like peristalsis, Kim et al. [258] developed a microfluidic “gut-on-a-chip” model that featured intestinal epithelium with villus-like structures. This system supported the growth of *L. rhamnosus* for over a week without harming the host cells. Because these models require only small volumes of cells and reagents, organoid-on-a-chip platforms can offer a more cost-effective and faster option for screening, although they do require technically complex and expensive equipment to set up [259]. Cells cultured within these microfluidic systems can be harvested for omics analyses, such as transcriptomics [260], while embedded microsensors allow for real-time monitoring of cellular events—like biofilm formation, barrier integrity, cell migration, protein secretion, and fluid dynamics [261]. Organoids have been used to explore the preferred infection sites of *S. typhimurium* and *L. monocytogenes*, and a triple culture of organoids, with *L. casei*, and *Bifidobacterium longum*, can stimulate organoid barrier formation and increased mucin production and can be used to study epithelial infection by a variety of microorganisms [262].

Though still a relatively recent innovation, organoid-on-a-chip systems are extremely versatile and may be among the most biologically realistic models available that still support efficient, large-scale testing. Each model type—whether ALI, spheroid, or chip-based—has its pros and cons, especially regarding cost, ease of use, and resemblance to in vivo biofilms. Of these, the physiological similarity to human infections is likely the most crucial factor in developing effective anti-biofilm therapies. These models can also be paired with other types of analysis, such as omics approaches, to deepen our understanding of drug mechanisms and microbial behavior.

## 7. Conclusions

The integration of advanced technologies—ranging from optical imaging and genetic tools to microfluidics and organoid models—has significantly enhanced our understanding of microbial biofilms in food-related environments. These tools have not only improved detection and characterization of biofilms but also enabled the identification of novel anti-biofilm agents and microbial interactions. However, the successful translation of these technologies into real-world food production settings remains a challenge.

Future research should focus on or improve some key areas: (1) the validation of these approaches in industrial-scale conditions, particularly under dynamic processing environments; (2) the integration of multi-omics data (genomics, transcriptomics, metabolomics) with real-time biosensors to enable predictive and preventive interventions; (3) the design of strain-specific and matrix-targeted sanitation protocols, informed by the mechanical and biochemical properties of biofilms; and (4) the development of eco-friendly biofilm control strategies, such as quorum sensing inhibitors, enzyme-based treatments, and biopreservative consortia.

Addressing these challenges will not only improve food safety and quality but also support sustainable and innovative food processing solutions tailored to the microbial ecology of each production context.

## Data Availability

Not applicable.

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
