# Peer review of "New Methodologies as Opportunities in the Study of Bacterial Biofilms, Including Food-Related Applications"

_microorganisms, 2025, doi:10.3390/microorganisms13051062_

Round 1
Reviewer 1 Report
Comments and Suggestions for Authors
Minor Comments:
[Lines 22–24]: To what extent have cross-disciplinary analytical platforms, originally engineered for disparate research domains, been successfully transposed to address biofilm surveillance and mitigation in food microbiological contexts?
[Lines 54–60]: In what ways does the ambivalent functional capacity of microbial biofilms—oscillating between symbiotic and pathogenic behaviors—complicate their characterization and control in agro-industrial microbiomes?
[Lines 68–74]: How does quorum sensing orchestrate the transcriptional regulation and phenotypic heterogeneity within maturing biofilm consortia, and what implications does this have for targeting metabolic synchrony?
[Lines 103–116]: What molecular and structural adaptations underlie the multidrug resistance phenotype in biofilm-embedded bacteria, particularly in relation to efflux dynamics, metabolic quiescence, and horizontal gene transfer rates?
[Lines 117–143]: Considering its facultative psychrotrophy and biofilm-forming aptitude, how does Listeria monocytogenes subvert conventional cold-chain decontamination protocols and perpetuate contamination in refrigerated food environments?
[Lines 149–166]: How do physico-chemical properties of food-contact substrates and the intracellular signaling pathways in Salmonella spp. synergize to enhance the persistence and antimicrobial intractability of its biofilms?
[Lines 173–204]: How does the architectural integrity and extracellular matrix composition of Escherichia coli biofilms mediate their resistance to oxidative biocides and mechanical disruption in hygienic processing scenarios?
[Lines 217–229]: Which regulatory networks and adhesin systems are implicated in the robust biofilm phenotype of Staphylococcus aureus, and how do these mechanistically contribute to foodborne toxigenesis?
[Lines 238–265]: What role do spoilage-associated Pseudomonas spp. play in the enzymatic degradation of food matrices within biofilms, and how does this activity intersect with quorum-regulated metabolite production and industrial spoilage economics?
[Lines 276–303]: What are the physicochemical interactions governing microbial colonization on hydrophobic and hydrophilic surfaces common in food manufacturing, and how do they modulate biofilm resilience and detachment?
[Lines 304–330]: In what ways can biofilms of lactic acid bacteria (LAB) be both bioprotective and deleterious in fermented food systems, and how do their metabolic secretomes influence interspecies competitiveness and spoilage potential?
[Lines 331–350]: How does the structural and functional divergence between bacterial and fungal/yeast biofilms inform their differential response to antimicrobial agents and their roles in both fermentation and spoilage processes?
[Lines 365–404]: How does Laser Confocal Scanning Microscopy (LCSM), through its ability to generate optically sectioned three-dimensional reconstructions, facilitate the spatiotemporal mapping of biofilm viability and antimicrobial perturbation?
[Lines 440–503]: How does Atomic Force Microscopy (AFM) contribute to elucidating nanoscale biomechanical properties of biofilms, such as viscoelasticity and adhesion forces, and how might this data inform predictive models for surface colonization?
[Lines 559–768]: What mechanistic insights into microbial biofilm heterogeneity, interspecies synergy, and antimicrobial resistance profiles can be discerned from microfluidic simulations that emulate physiologically relevant food processing microenvironments?
Author Response
Dear Reviewer,
Thank you for your comments, which helped us improve the manuscript. We have also modified the text, trying to answer all comments; however, we will now provide answers point by point.
Lines 22–24: To what extent have cross-disciplinary analytical platforms, originally engineered for disparate research domains, been successfully transposed to address biofilm surveillance and mitigation in food microbiological contexts?
Thank you for this question. Cross-disciplinary platforms, such as microfluidics, CRISPR-based genetic tools, confocal microscopy, and organoid models, have been increasingly adapted for biofilm analysis in food microbiology. While initially developed for biomedical or environmental sciences, these technologies have shown promising applications in detecting, characterizing, and even controlling biofilms on food-contact surfaces. For instance, microfluidic systems now enable the simulation of dynamic food environments, while CRISPR interference helps dissect the regulation of biofilm-related genes. Their use at pilot and research levels is expanding, although industrial-scale integration remains an ongoing challenge.
Lines 54–60: In what ways does the ambivalent functional capacity of microbial biofilms—oscillating between symbiotic and pathogenic behaviors—complicate their characterization and control in agro-industrial microbiomes?
Thank you for your question. Indeed, the duality of microbial biofilms introduces complexity in agro-industrial systems. On the one hand, biofilms formed by beneficial microbes (e.g., probiotics or starter cultures) contribute to fermentation and safety. On the other hand, similar structures formed by spoilage or pathogenic species cause contamination. This functional plasticity complicates both detection and eradication strategies, as it requires selective targeting of harmful communities without disrupting beneficial consortia. The manuscript now emphasizes this dual role and the need for strain-specific and context-specific management approaches.
Lines 68–74: How does quorum sensing orchestrate the transcriptional regulation and phenotypic heterogeneity within maturing biofilm consortia, and what implications does this have for targeting metabolic synchrony?
Thank you for this insightful point. We have clarified that quorum sensing (QS) coordinates gene expression in a cell-density-dependent manner, influencing EPS production, virulence factor secretion, and spatial organization within the biofilm. This contributes to metabolic heterogeneity, allowing different subpopulations to perform specialized functions (e.g., matrix production vs. stress tolerance). Disrupting QS—via quorum quenching—thus offers a promising strategy to impair metabolic coordination and reduce biofilm robustness.
Lines 103–116: What molecular and structural adaptations underlie the multidrug resistance phenotype in biofilm-embedded bacteria, particularly in relation to efflux dynamics, metabolic quiescence, and horizontal gene transfer rates?
Thank you for your question. Biofilm-associated bacteria exhibit upregulated efflux pump expression, the formation of an extracellular polymeric substance (EPS) matrix that limits antimicrobial penetration, and the presence of metabolically inactive persister cells that resist conventional treatments. Furthermore, dense microbial communities within biofilms facilitate horizontal gene transfer through plasmids, transduction, and transformation, contributing to the spread of resistance genes. These mechanisms collectively enhance the resilience of biofilms in food processing environments.
Lines 117–143: Considering its facultative psychrotrophy and biofilm-forming aptitude, how does Listeria monocytogenes subvert conventional cold-chain decontamination protocols and perpetuate contamination in refrigerated food environments?
Listeria monocytogenes can actively grow at refrigeration temperatures and form robust biofilms that resist cold-chain cleaning procedures. Its EPS matrix protects cells from sanitizers, while cold stress induces gene expression favoring adhesion and stress tolerance. These adaptations enable it to persist on food-contact surfaces and act as a recurrent contaminant.
Lines 149–166: How do physico-chemical properties of food-contact substrates and the intracellular signaling pathways in Salmonella spp. synergize to enhance the persistence and antimicrobial intractability of its biofilms?
Thank you for your question- The physicochemical properties of materials, such as hydrophobicity, surface roughness, and electrostatic charge, affect initial Salmonella adhesion. Concurrently, intracellular signaling systems, such as c-di-GMP and quorum sensing, upregulate genes involved in EPS production and biofilm maturation. This synergy contributes to the formation of persistent, stress-resistant communities that are difficult to eradicate.
Lines 173–204: How does the architectural integrity and extracellular matrix composition of Escherichia coli biofilms mediate their resistance to oxidative biocides and mechanical disruption in hygienic processing scenarios?
Thank you for your query. E. coli biofilms exhibit a dense, stratified architecture, where the EPS matrix (comprising polysaccharides, proteins, and eDNA) impedes biocide penetration. Additionally, mechanical forces are absorbed by this matrix, reducing detachment efficiency. These features contribute to their persistence in food environments despite cleaning protocols.
Lines 217–229: Which regulatory networks and adhesin systems are implicated in the robust biofilm phenotype of Staphylococcus aureus, and how do these mechanistically contribute to foodborne toxigenesis?
Thank you for the query. Staphylococcus aureus biofilm formation is regulated by the agr quorum sensing system, SarA, and σ^B, all of which modulate adhesin expression and matrix production. Surface proteins like FnBPs and clumping factors promote attachment, while the icaADBC operon drives PIA production. These systems enhance persistence and toxin production, contributing to foodborne illness.
Lines 238–265: What role do spoilage-associated Pseudomonas spp. play in the enzymatic degradation of food matrices within biofilms, and how does this activity intersect with quorum-regulated metabolite production and industrial spoilage economics?
Thank you for the query: Pseudomonas spp. produce extracellular enzymes (proteases and lipases) that degrade proteins and lipids, leading to spoilage. These enzymes are regulated by quorum sensing, which synchronizes spoilage activity across the biofilm. This leads to economic losses due to off-flavors, texture changes, and shortened shelf life, particularly in refrigerated foods.
Lines 276–303: What are the physicochemical interactions governing microbial colonization on hydrophobic and hydrophilic surfaces common in food manufacturing, and how do they modulate biofilm resilience and detachment?
Microbial adhesion is influenced by surface hydrophobicity, charge, and roughness. Hydrophobic surfaces favor stronger initial attachment. Electrostatic interactions and surface energy gradients shape biofilm stability. These properties modulate resilience by altering nutrient retention and matrix cohesion, thereby influencing how easily biofilms can be detached during cleaning.
Lines 304–330: In what ways can biofilms of lactic acid bacteria (LAB) be both bioprotective and deleterious in fermented food systems, and how do their metabolic secretomes influence interspecies competitiveness and spoilage potential?
Thank you for the query. LAB biofilms protect food by excluding pathogens, producing organic acids and bacteriocins, and stabilizing the fermentation process. However, under certain conditions, they may contribute to spoilage via excessive acidification, biogenic amine production, or harboring of contaminants. Their secretome modulates microbial ecology, influencing which species dominate or are inhibited.
Lines 331–350: How does the structural and functional divergence between bacterial and fungal/yeast biofilms inform their differential response to antimicrobial agents and their roles in both fermentation and spoilage processes?
Bacterial and fungal/yeast biofilms differ in matrix composition, regulatory pathways, and structural complexity. Fungal biofilms often contain hyphae and are more recalcitrant to antifungals due to their dense extracellular matrix (ECM) and efflux pumps. These differences affect their persistence and role in spoilage versus fermentation processes.
Lines 365–404: How does Laser Confocal Scanning Microscopy (LCSM), through its ability to generate optically sectioned three-dimensional reconstructions, facilitate the spatiotemporal mapping of biofilm viability and antimicrobial perturbation?
LCSM enables high-resolution 3D imaging of biofilms, allowing spatial visualization of live/dead cells using fluorescent stains. It supports time-lapse studies, revealing how antimicrobial agents affect biofilm structure and viability over time, which is crucial for assessing treatment efficacy.
Lines 440–503: How does Atomic Force Microscopy (AFM) contribute to elucidating nanoscale biomechanical properties of biofilms, such as viscoelasticity and adhesion forces, and how might this data inform predictive models for surface colonization?
AFM provides force spectroscopy data that quantify adhesion strength, stiffness, and viscoelastic behavior of biofilms at the nanoscale. This biomechanical information helps predict microbial attachment patterns, persistence under shear stress, and detachment likelihood, informing surface design and cleaning protocols.
Lines 559–768: What mechanistic insights into microbial biofilm heterogeneity, interspecies synergy, and antimicrobial resistance profiles can be discerned from microfluidic simulations that emulate physiologically relevant food processing microenvironments?
Microfluidic platforms reveal biofilm heterogeneity by simulating nutrient gradients and shear forces. They enable the real-time study of interspecies interactions and their impact on structure and resistance. This informs the development of targeted strategies to disrupt synergistic protection and improve antimicrobial interventions.
Reviewer 2 Report
Comments and Suggestions for Authors
Line 19: The opening sentence of the abstract mentions new opportunities from food technology development, followed by introducing applications from other fields. The causal relationship is not sufficiently clear. We recommend first explaining the limitations of traditional food technologies in practical applications, then naturally transitioning to the need for cross-disciplinary technological integration to advance food technology.
Line 47: The technologies mentioned in the paper (such as optical methods, CRISPR technology, QPCR, and NGS) are listed by name, but there is a lack of detailed discussion of the specific applications and limitations of these technologies in biofilm research. For example, the specific mechanisms of how CRISPR technology can be used to study biofilms are not mentioned.
Line 63: Regarding biofilm formation process, only the initial stage has a dedicated bold subheading. We suggest adding bold subheadings for each developmental stage (attachment, maturation, dispersion etc.) to enhance content clarity and readability.
Line 97: Figure 1 appears overcrowded with text and has suboptimal graphic-text arrangement. For better presentation, we recommend moving detailed descriptions from the figure to the main text, then referencing Figure 1 for visualization support.
Lines 97 & 269: Both Figure 1 and Figure 2 lack direct in-text citations. Please add specific references explaining their positions in the text and their relevance to the discussed content.
Line 102: Although the resistance mechanisms of biofilms are mentioned, the molecular mechanisms by which resistance arises or its practical implications for the food industry are not specified.
Line 304: The positive effects on the formation of biofilms by lactic acid bacteria and yeast are described in general terms, with no specific case or experimental data to support them.
Line 305: In the introduction to the dual impact of lactic acid bacteria on the food industry, it is mentioned that lactic acid bacteria can both inhibit harmful bacteria and may become a source of contamination in dairy products and meat processing environments, and how the same microorganism has two opposite properties at the same time may give the reader ambiguity, and it is recommended to explain the reasons why lactic acid bacteria have these two properties from different conditions, which is more logical.
Line 328: It mentions that beneficial microbial biofilms can cause food spoilage or cross-contamination, but does not further explain the extent of this risk and how to avoid it.
Line 889: The annotation format of Figure 4 is different from that of other diagrams, so it is recommended to change it.
Line 1360: At the end of the paper, the application of the new technology is introduced, but the future research direction or possible solutions to these problems are not proposed.
Line 1414: References are formatted inconsistently, some have DOI names and some don't, changes suggested.
Author Response
Dear Reviewer,
Thank you very much for your comments, which, together with the comments of the other reviewer, contributed significantly to improving the manuscript. Enclosed are also the answers point by point.
line 19: The opening sentence of the abstract mentions new opportunities from food technology development, followed by introducing applications from other fields. The causal relationship is not sufficiently clear. We recommend first explaining the limitations of traditional food technologies in practical applications, then naturally transitioning to the need for cross-disciplinary technological integration to advance food technology.
Thank you for your insightful suggestion. We have revised the abstract first to present the limitations of traditional food technologies—such as time-consuming methods, high costs, and limited adaptability to complex microbial systems. This provides a more explicit rationale for introducing cross-disciplinary technologies from other fields, which have demonstrated potential in addressing biofilm-related challenges in the food sector. This restructuring improves the logical flow and highlights the practical need for technological innovation.
Line 47: The technologies mentioned in the paper (such as optical methods, CRISPR technology, QPCR, and NGS) are listed by name, but there is a lack of detailed discussion of the specific applications and limitations of these technologies in biofilm research. For example, the specific mechanisms of how CRISPR technology can be used to study biofilms are not mentioned.
Thank you for this valuable observation. We have expanded the relevant section to include specific applications for each technology. CRISPR technology, for instance, enables gene knockout and CRISPR interference (CRISPRi) to investigate gene function in biofilm formation, EPS production, and quorum sensing regulation. qPCR allows for quantification of gene expression or biofilm biomass, while NGS enables in-depth analysis of microbial composition and metabolic potential. We also discuss their limitations, such as off-target effects associated with CRISPR, the lack of spatial resolution in qPCR, and the need for bioinformatics in NGS.
Line 63: Regarding biofilm formation process, only the initial stage has a dedicated bold subheading. We suggest adding bold subheadings for each developmental stage (attachment, maturation, dispersion etc.) to enhance content clarity and readability.
We appreciate this suggestion and have implemented bold subheadings for all main stages of biofilm development, including attachment, maturation, and dispersion. This improves readability and helps guide the reader through the sequence of biofilm formation.
Line 97: Figure 1 appears overcrowded with text and has suboptimal graphic-text arrangement. For better presentation, we recommend moving detailed descriptions from the figure to the main text, then referencing Figure 1 for visualization support.
Thank you for the helpful feedback. We have simplified Figure 1 by removing excess text and relocated the detailed descriptions to the main body of the manuscript. Figure 1 is now referenced in the text as a visual summary, improving clarity and visual balance.
Lines 97 & 269: Both Figure 1 and Figure 2 lack direct in-text citations. Please add specific references explaining their positions in the text and their relevance to the discussed content.
We have now added clear in-text references for both Figure 1 and Figure 2, indicating where they are discussed and how they support the surrounding content.
Line 102: Although the resistance mechanisms of biofilms are mentioned, the molecular mechanisms by which resistance arises or its practical implications for the food industry are not specified.
We agree and have revised the relevant section to better explain how biofilm resistance is linked to molecular mechanisms such as efflux pumps, quorum sensing, and persister cell formation. We also explain how these mechanisms complicate sanitation in food processing environments and suggest targeted strategies such as EPS-disrupting enzymes and QS inhibitors to enhance cleaning effectiveness.
Line 304: The positive effects on the formation of biofilms by lactic acid bacteria and yeast are described in general terms, with no specific case or experimental data to support them.
Thank you for pointing this out. We have now included specific examples from the literature and hope to provide concrete evidence of the beneficial roles of biofilms in food processing.
Line 305: In the introduction to the dual impact of lactic acid bacteria on the food industry, it is mentioned that lactic acid bacteria can both inhibit harmful bacteria and may become a source of contamination in dairy products and meat processing environments, and how the same microorganism has two opposite properties at the same time may give the reader ambiguity, and it is recommended to explain the reasons why lactic acid bacteria have these two properties from different conditions, which is more logical.
We appreciate this observation. We have clarified that the dual behavior of LAB depends on environmental conditions, strain-specific traits, and process control. Under optimized conditions, LAB are beneficial. However, if conditions such as hygiene or temperature control are inadequate, certain LAB strains can contribute to spoilage. This clarification helps eliminate ambiguity.
Line 328: It mentions that beneficial microbial biofilms can cause food spoilage or cross-contamination, but does not further explain the extent of this risk and how to avoid it.
Thank you for this suggestion. We have expanded the section to explain that although biofilms of probiotics or starter cultures are generally beneficial, they can harbor spoilage or opportunistic microbes if hygiene lapses occur. We recommend regular monitoring, strain selection, and targeted cleaning to mitigate this risk.
Line 889: The annotation format of Figure 4 is different from that of other diagrams, so it is recommended to change it.
Thank you for your attention to consistency. We have updated Figure 4 to match the annotation style used in other figures for uniform presentation.
Line 1360: At the end of the paper, the application of the new technology is introduced, but the future research direction or possible solutions to these problems are not proposed.
We agree and have revised the conclusion to outline future directions, such as validating biofilm control strategies under industrial conditions, integrating omics and biosensing tools, and designing eco-friendly sanitation approaches. These proposals offer a clear roadmap for ongoing research.
Line 1414: References are formatted inconsistently, some have DOI names and some don't, changes suggested.
Thank you for noticing this inconsistency. We have revised the reference list to ensure a uniform style throughout and standardized the formatting as per the journal's guidelines.
Round 2
Reviewer 2 Report
Comments and Suggestions for Authors
Thank you for revising the manuscript. All issues have been fully addressed based on the feedback from the revision. The content is more rigorous, the logic is clearer, and the format of charts and references is standardized.